 RESEARCH ADVANCE

# Investigating macroecological patterns in coarse-grained microbial communities using the stochastic logistic model of growth

**William R Shoemaker*, Jacopo Grilli**

Quantitative Life Sciences, The Abdus Salam International Centre for Theoretical Physics (ICTP), Trieste, Italy

**\*For correspondence:**
williamrshoemaker@gmail.com

**Competing interest:** The authors declare that no competing interests exist.

**Abstract** The structure and diversity of microbial communities are intrinsically hierarchical due to the shared evolutionary history of their constituents. This history is typically captured through taxonomic assignment and phylogenetic reconstruction, sources of information that are frequently used to group microbes into higher levels of organization in experimental and natural communities. Connecting community diversity to the joint ecological dynamics of the abundances of these groups is a central problem of community ecology. However, how microbial diversity depends on the scale of observation at which groups are defined has never been systematically examined. Here, we used a macroecological approach to quantitatively characterize the structure and diversity of microbial communities among disparate environments across taxonomic and phylogenetic scales. We found that measures of biodiversity at a given scale can be consistently predicted using a minimal model of ecology, the Stochastic Logistic Model of growth (SLM). This result suggests that the SLM is a more appropriate null-model for microbial biodiversity than alternatives such as the Unified Neutral Theory of Biodiversity. Extending these within-scale results, we examined the relationship between measures of biodiversity calculated at different scales (e.g. genus vs. family), an empirical pattern previously evaluated in the context of the Diversity Begets Diversity (DBD) hypothesis (Madi et al., 2020). We found that the relationship between richness estimates at different scales can be quantitatively predicted assuming independence among community members, demonstrating that the DBD can be sufficiently explained using the SLM as a null model of ecology. Contrastingly, only by including correlations between the abundances of community members (e.g. as the consequence of interactions) can we predict the relationship between estimates of diversity at different scales. The results of this study characterize novel microbial patterns across scales of organization and establish a sharp demarcation between recently proposed macroecological patterns that are not and are affected by ecological interactions.

## eLife assessment

This **valuable** study considers empirical macroecological patterns in microbiome data across multiple taxonomic scales. The work **convincingly** shows that the Stochastic Logistic Growth model is a more appropriate choice of null model than the neutral theory of biodiversity. The work will be of particular interest to microbial ecologists.

## Introduction

An essential feature of microbial communities is their heterogeneous composition. A single environmental sample typically has a high richness, harboring hundreds to thousands of community members (*Thompson et al., 2017*; *Shoemaker et al., 2017*; *Barberán et al., 2014*). This high level of richness reaches an astronomical quantity at the global level, as scaling relationships and models of biodiversity predict upwards of one trillion (~$10^{12}$) species on Earth (*Locey and Lennon, 2016*; *Lennon and Locey, 2020*). Even experimental communities in laboratory settings with a single carbon source can harbor ≥40 community members, culminating in a total richness numbering in the hundreds among replicate communities (e.g. *Dal Bello et al., 2021*). This richness contributes to the sheer diversity of microbial communities, a challenge for researchers attempting to identify the general principles that govern their dynamics and composition.

While richness estimates of microbial communities are undoubtedly high, the choice of assigning a community member to a given taxon remains intrinsically arbitrary. This arbitrariness exists regardless of whether the definition of a taxon is based on physiological attributes measured in the laboratory, entire genomes (i.e. metagenomics), or single-gene amplicon-based methods (i.e. 16S rRNA annotation). Despite their methodological differences, these approaches can all be viewed as different ways to cluster individuals within a community into groups. To contend with the sheer richness of microbial communities, researchers frequently rely on annotation-based approaches, i.e., by summing the abundances of community members that belong to the same group at a given taxonomic scale (e.g. genus, family, etc.). This approach pares down communities to a size that is amenable for the visualization of individual groups and allows for questions of scale-dependent community reproducibility to be addressed (*Louca et al., 2016*; *Goldford et al., 2018*; *Estrela et al., 2022*; *Estrela et al., 2021*; *Dal Bello et al., 2021*; *Ho et al., 2022*; *Good and Rosenfeld, 2022*; *Tian et al., 2020*).

This movement towards performing analyses of diversity at various taxonomic scales raises the question of how the composition of a community at one scale relates to that at another. To address these questions, researchers have examined the relationship between biodiversity measures at different scales in order to pare down the set of ecological mechanisms that plausibly govern community composition. Specifically, recent efforts have found that microbial richness/diversity within a given taxonomic group (e.g. genus) is typically positively correlated with the richness/diversity among the remaining groups (e.g. family) (*Madi et al., 2020*; *Estrela et al., 2022*), an empirical pattern that aligns with the predictions of the Diversity Begets Diversity (DBD) hypothesis (*Whittaker, 1972*; *San Roman et al., 2018*; *Maynard et al., 2017*). Evidence of the DBD hypothesis has historically been attributed to the construction of novel niches within a community through member interactions (*Calcagno et al., 2017*; *Whittaker, 1972*), with similar mechanisms having been proposed to explain the existence of a positive relationship in microbial communities (*Madi et al., 2020*). However, we still lack a quantitative understanding of how community composition at one scale should relate to that of another. Proceeding towards this goal requires two elements: (1) a systematic approach to grouping community members and (2) an appropriate null model for the composition of communities.

The operation of grouping the components of a system into a smaller number (e.g. merging read counts of OTUs to the family level in a community) is known in the physical sciences as coarse-graining. This formalism defines our systematic approach to grouping community members. While it is often not explicitly acknowledged as such, coarse-graining is a core concept in the microbial life sciences (*Good and Hallatschek, 2018*). By smoothing over microscopic details at a lower level of biological organization in order to make progress at a higher level, the concept of coarse-graining has contributed towards the development of effective models of physiological growth (*Scott et al., 2014*; *Jun et al., 2018*), evolutionary dynamics (*Schweinsberg, 2003*; *Desai et al., 2013*), and the dependence of ecosystem properties on the diversity of underlying communities (*Moran and Tikhonov, 2022*). Coarse-graining has even been used to glean insight into the question of whether 'species' as a unit has meaning for microorganisms, as modeling efforts have found that the operation permits the delimitation of distinct taxonomic groups when the resource preferences of community members are structured (*Tikhonov, 2017*). These theoretical and empirical efforts suggest that coarse-graining may provide an appropriate framework for investigating patterns of diversity and abundance within and between taxonomic scales of observation.

When evaluating the novelty of an empirical pattern it is useful to identify an appropriate null model for comparison (*O'Dwyer et al., 2017*; *McGill, 2010*; *Harte, 2011*). Prior research efforts

have demonstrated the novelty of the fine vs. coarse-grained relationship by contrasting inferences from empirical data with predictions obtained from the Unified Neutral Theory of Biodiversity (UNTB) (*Hubbell, 2011*; *Volkov et al., 2003*; *Azaele et al., 2016*; *Madi et al., 2020*; *Alonso and McKane, 2004*; *Azaele et al., 2006*). These predictions generally failed to reproduce slopes inferred from empirical data (*Madi et al., 2020*), implying that the fine vs. coarse-grained relationship represents a novel macroecological pattern that cannot be quantitatively explained by existing null models of ecology. However, the task of identifying an appropriate null model for comparison is not straightforward. Rather, the question of what constitutes an appropriate null model remains a persistent topic of discussion in community ecology (*Simberloff, 1983*; *Harvey et al., 1983*; *Gotelli and Graves, 1996*; *Gotelli and Ulrich, 2012*; *O'Dwyer et al., 2017*). Here, we take the view that a null model is appropriate for examining the relationship between two observables (e.g. community diversity at different scales) if it was capable of quantitatively predicting each observable (e.g. community diversity at one scale). By this standard, the UNTB is an unsuitable choice as a null as it generally fails to capture basic patterns of microbial diversity and abundance at any scale (*Li and Ma, 2016*; *Harris et al., 2017*; *Grilli, 2020*). One relevant example is that the UNTB predicts that the distribution of mean abundances of community members across sites is extremely narrow (i.e. converging to a delta distribution as the number of sites increases), whereas empirical data tends to follow a broad lognormal distribution (*Grilli, 2020*). Contrastingly, recent efforts have determined that the predictions of a model of self-limiting growth with environmental noise, the SLM, is capable of quantitatively capturing multiple empirical macroecological patterns in observational and experimental microbial communities (*Grilli, 2020*; *Zaoli and Grilli, 2021*; *Zaoli et al., 2022*; *Descheemaeker et al., 2021*; *Descheemaeker and de Buyl, 2020*; *Shoemaker et al., 2023c*; *Lim et al., 2023*). The stationary solution of this model predicts that the abundance of a given community member across sites follows a gamma distribution (*Grilli, 2020*), a result that provides the foundation necessary to predict macroecological patterns among and between different taxonomic and phylogenetic scales.

In this study, we evaluated macroecological patterns of microbial communities across scales of evolutionary resolution. To limit potential biases that may result due to taxonomic annotation errors and to use all available data, we investigated the macroecological consequences of coarse-graining by developing a procedure that groups community members using the underlying phylogeny in addition to relying on taxonomic assignment. We used data from the Earth Microbiome Project (EMP), a public catalog of microbial community barcode data, to ensure the generality of our findings and their commensurability with past research efforts. First, we assessed the extent that microbial diversity varies as the abundances of community members are coarse-grained by phylogenetic distance and taxonomic rank. The results of these analyses lead us to consider whether the predictive capacity of the gamma distribution remained robust under coarse-graining, a prediction that we quantitatively evaluated among community members and then extended to predict overall community richness and diversity. The accuracy of the gamma distribution provided the necessary motivation to test whether the gamma distribution was capable of predicting the relationship between fine and coarse-grained estimates, the empirical pattern that has been interpreted as evidence for the DBD hypothesis. Together, these analyses present evidence of the scale invariance of macroecological patterns in microbial communities as well as the applicability of the gamma distribution, the stationary distribution of the SLM, as a null model for evaluating the novelty of macroecological patterns of microbial biodiversity.

## Results
### The macroecological consequences of phylogenetic and taxonomic coarse-graining

While microbial communities are often coarse-grained into higher taxonomic scales, their effect on measures of biodiversity and the underlying phylogeny are rarely examined. Before proceeding with the full analysis using public 16S rRNA amplicon data from the EMP, we elected to quantify the fraction of the remaining community members across coarse-graining thresholds, a reflection of the extent that coarse-graining reduces global richness and the relation between taxonomic and phylogenetic coarse-graining. We first defined a coarse-grained group $g$ as the set of OTUs that have the same assigned label in a given taxonomic rank out of $G$ groups (e.g. *Pseudomonas* at the genus level) or are

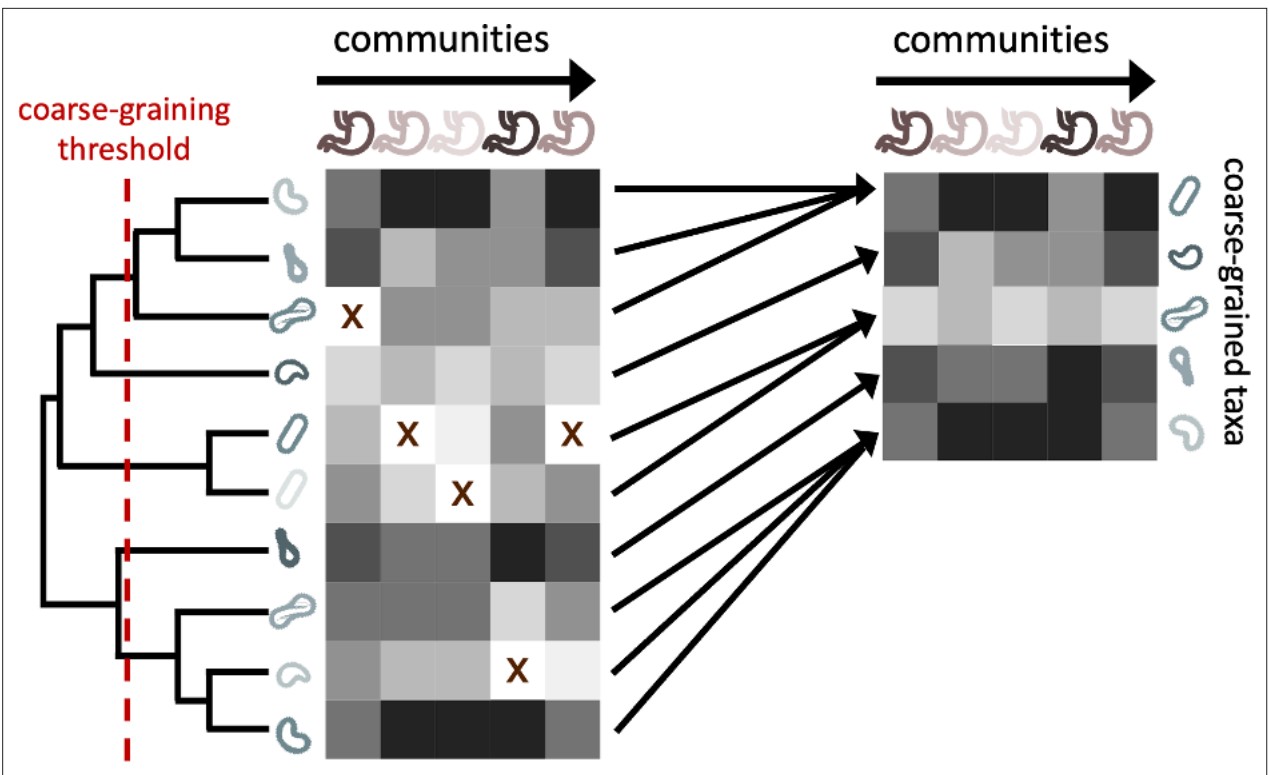

**Figure 1.** The process of coarse-graining abundances using the phylogeny. Taxonomic assignment in 16S rRNA amplicon sequence data provided the opportunity to investigate how properties of communities vary at different taxonomic scales. The most straightforward means of coarse-graining here is to sum the abundances of OTUs/ASVs that belong to the same taxonomic group. Amplicon data-based studies provide information about the shared evolutionary history of community constituents, information that can be leveraged by the construction of phylogenetic trees. A coarse-graining procedure can be defined that is analogous to one based on taxonomy, where a phylogenetic root-to-tip distance is chosen and terminal nodes are collapsed if their distance to a common ancestor is less than the prescribed distance.

The online version of this article includes the following figure supplement(s) for figure 1:

**Figure supplement 1.** The process of coarse-graining using taxonomic information.

**Figure supplement 2.** Examining the change in relative richness under coarse-graining.

collapsed when the phylogeny is truncated by a given root-to-tip distance (**Figure 1**, **Figure 1—figure supplement 1**). The relative abundance of group $g$ in site $j$ is defined as $x_{g,j} = \sum_{i \in g} x_{i,j}$.

We found that even minor degrees of coarse-graining had a drastic effect on the total number of community members within an environment, reducing global richness by ~90% even at just the genus level (**Figure 1—figure supplement 2a**). By coarse-graining over a range of phylogenetic distances, we found that the fraction of coarse-grained community members comparable to that of genus-level coarse-graining occurred at a root-to-tip phylogenetic distance of ~0.1 (**Figure 1—figure supplement 2b**). This distance translated to only ~3% of the total distance of the tree, meaning that the majority of OTUs were coarse-grained over a minority of the tree. This pattern was likely driven by the underlying structure of microbial phylogenetic trees, where most community members have short branch lengths (**O'Dwyer et al., 2015**). This result suggests that while coarse-graining communities to the genus or family level substantially reduces global richness, it does so without coarse-graining the majority of the evolutionary history captured by the phylogeny. Assuming that phylogenies capture ecological changes that occur over evolutionary time, this detail implies that ecological divergence that is captured by the phylogeny should be retained even when communities are considerably coarse-grained.

With our coarse-graining procedures established, we proceeded with our macroecological investigation. Recent efforts have found that the distribution of abundances of a given ASV/OTU maintained a consistent statistically similar form across independent sites and time, a pattern known as the Abundance Fluctuation Distribution (**Grilli, 2020**; **Zaoli and Grilli, 2021**; **Zaoli et al., 2022**; **Shoemaker,**

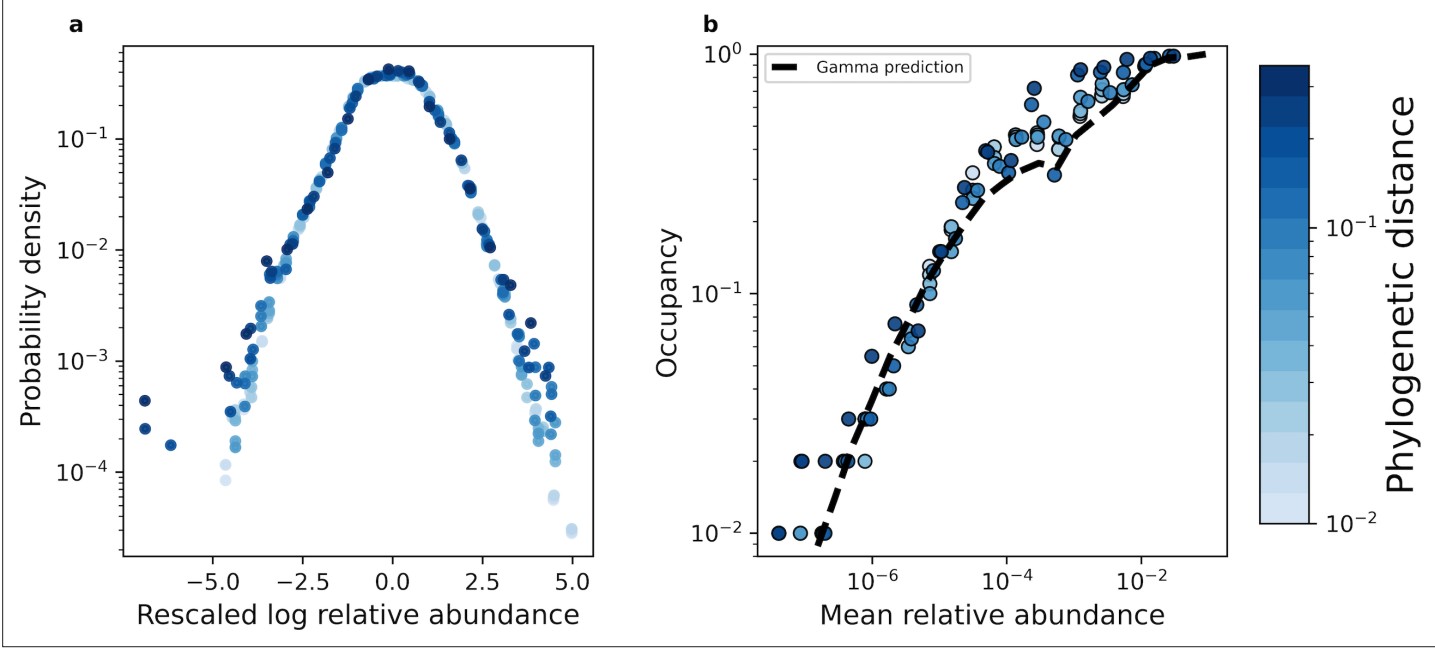

**Figure 2.** The shape of the AFD remained qualitatively invariant under coarse-graining. (**a**) Under phylogenetic coarse-graining the general shape of the AFD for OTUs that were present in all sites (i.e. an occupancy of one) remained qualitatively invariant. (**b**) Similarly, the shape of the relationship between the mean coarse-grained abundance across hosts and occupancy across sites did not tend to vary. Predictions obtained from the gamma distribution are capable of capturing the relationship between the mean abundance and occupancy, suggesting that the gamma distribution remains a useful quantitative null model under coarse-graining. All data in this plot is from the human gut microbiome.

The online version of this article includes the following figure supplement(s) for figure 2:

**Figure supplement 1.** The AFD of all environments under taxonomic coarse-graining.

**Figure supplement 2.** The AFD of all environments under phylogenetic coarse-graining.

**Figure supplement 3.** The predicted occupancy across sites for a gamma-distributed AFD under taxonomic coarse-graining for all environments.

**Figure supplement 4.** The predicted occupancy across sites for a gamma distributed AFD under phylogenetic coarse-graining for all environments.

**Figure supplement 5.** The relationship between the mean abundance across sites and the occupancy for various taxonomic coarse-graining scales.

**Figure supplement 6.** The relationship between the mean abundance across sites and the occupancy for various phylogenetic coarse-graining scales.

**Figure supplement 7.** Predictions of the variance of occupancy failed across taxonomic coarse-graining thresholds.

**Figure supplement 8.** Predictions of the variance of occupancy failed across phylogenetic coarse-graining thresholds.

**Figure supplement 9.** Occupancy predictions of the gamma remained invariant despite coarse-graining.

**Figure supplement 10.** The sum of the variances of OTUs was close to the value of the variance of a taxonomic coarse-grained group, implying that the contribution of covariance to the variance of a given coarse-grained group was low.

**Figure supplement 11.** The analysis presented in *Figure 2—figure supplement 10* but for phylogenetic coarse-graining.

**Figure supplement 12.** The plot presented in *Figure 2—figure supplement 10* but with the ratio of coarse and fine-grained variances plotted on the y-axis for the purpose of visualizing deviations from the 1:1 line.

**Figure supplement 13.** The analysis presented in *Figure 2—figure supplement 12* but for phylogenetic coarse-graining.

*2023a*; *Wolff et al., 2023*). By coarse-graining empirical AFDs and rescaling them by their mean and variance across sites (i.e. standard score), we found that AFDs from the human gut microbiome retained their shape across phylogenetic scales (*Figure 2a*). This pattern of invariance held across environments for both phylogenetic and taxonomic coarse-graining (*Figure 2—figure supplement 1*, *Figure 2—figure supplement 2*), suggesting that empirical AFDs can likely be described by a single probability distribution.

It has been previously demonstrated that empirical microbial AFDs are well-described by a gamma distribution that is parameterized by the mean relative abundance $\bar{x}_i$ and the shape parameter $\beta_i = \frac{\bar{x}_i^2}{\sigma_i^2}$ (equal to the squared inverse of the coefficient of variation *Grilli, 2020*). This distribution can be

viewed as the stationary distribution of a SLM of growth, a mathematical model that successfully captures macroecological patterns of microbial communities across both sites and time (*Grilli, 2020*; *Descheemaeker and de Buyl, 2020*; *Zaoli and Grilli, 2021*; *Equation 5* in Materials and methods).

Using this result, we determined whether the gamma distribution sufficiently characterized coarse-grained AFDs. In order to accomplish this task, it is worth noting that we do not directly observe $x_i$. Rather, our ability to observe a community member is dependent on sampling effort (i.e. total number of reads for a given site). To account for sampling, one can derive a form of the gamma distribution that explicitly accounts for the sampling process, obtaining the probability of obtaining $n$ reads out of $N$ total reads belonging to a community member (Materials and methods, *Grilli, 2020*). Given that $n = 0$ for a community member, we do not observe, we defined the fraction of $M$ sites where a community member was observed (i.e. occupancy, $o_i$) as

$$\langle o_i \rangle = 1 - \frac{1}{M} \sum_{m=1}^{M} P(0|N_m, \bar{x}_i, \beta_i) \tag{1}$$

We then compared this prediction to observed estimates of occupancy to assess the accuracy of the gamma distribution across coarse-grained thresholds. We found that *Equation 1* generally succeeded in predicting observed occupancy across phylogenetic and taxonomic scales for all environments (*Figure 2—figure supplement 3*, *Figure 2—figure supplement 4*). We then determined whether the gamma distribution was capable of predicting the relationship between macroecological quantities. One such relationship is that the occupancy of a community member should increase with its mean abundance, known as the abundance-occupancy relationship (*Gaston et al., 2000*). This pattern has been found across microbial systems (*Shade et al., 2018*; *Sloan et al., 2007*; *Burns et al., 2016*) and can be quantitatively predicted using the gamma distribution (*Grilli, 2020*). We see that this relationship is broadly captured across taxonomic and phylogenetic scales for all environments (*Figure 2b*, *Figure 2—figure supplement 5*, *Figure 2—figure supplement 6*). This result implies that the ability to observe a given taxonomic group was primarily determined by its mean abundance across sites and the sampling effort within a site, regardless of one's scale of observation. In contrast, under the assumption of demographic indistinguishability under the UNTB we would expect the mean abundance distribution to be extremely narrow, following a delta distribution. Under the SLM, the variation in mean relative abundances we observed implies that the carrying capacities of community members vary over multiple orders of magnitude. We also note that at high mean abundances our predictions show slight variation, which is likely driven by variation in the shape parameter $\beta$ (*Figure 2b*). In contrast with these results, the gamma distribution was unable to predict the variance of occupancy under both taxonomic and phylogenetic coarse-graining (*Equation 11*; *Figure 2—figure supplement 7*, *Figure 2—figure supplement 8*), the implications of which we will address in a later section.

To quantitatively assess the accuracy of the gamma distribution we calculated the relative error of our mean occupancy predictions (*Equation 12*) for all coarse-graining thresholds. We found that the mean logarithm of the error only slightly increased for the initial taxonomic and phylogenetic scales, where it then exhibited a sharp decrease across environments (*Figure 2—figure supplement 9*). The error then only began to decrease once the community became highly coarse-grained, harboring a global richness (union of all community members in all sites for a given environment) <20. This result means that, if anything, the accuracy of the gamma distribution only improved with coarse-graining.

## Reconciling coarse-graining and the predictions of the gamma distribution

The consistent predictive success of the gamma distribution under coarse-graining raises the question of why it remains a sufficient null model. The sum of independent gamma-distributed random variables only returns a gamma through analytic calculation if all random variables have identical rate parameters ($\beta_i/\bar{x}_i = \beta/\bar{x}$), a requirement that microbial communities clearly do not meet since they typically harbor broad mean abundance distributions. Given that, a gamma AFD cannot predict the distribution of correlations between AFDs (*Grilli, 2020*), it is first worth examining whether the degree of dependence between AFDs shapes coarse-grained variables. We first consider the relation between the variance of the sum and the sum of variances.

$$\text{Var}(\sum_i^{S_{\text{tot}}} x_i) = \sum_i^{S_{\text{tot}}} \text{Var}(x_i) + 2 \sum_{i<j} \text{Cov}(x_i, x_j) \tag{2}$$

By plotting $\text{Var}(\sum_i^{S_{\text{tot}}} x_i)$ against $\sum_i^{S_{\text{tot}}} \text{Var}(x_i)$ across coarse-grained thresholds, we found that the contribution of covariance to individual coarse-grained taxa was weak, suggesting that the statistical moments at higher scales can be approximated by those at lower scales (*Figure 2—figure supplement 10*, *Figure 2—figure supplement 11*). Similar conclusions can be drawn by plotting the variances as a ratio, with slight deviations above a ratio of one, suggesting that coarse-grained variance was slightly higher (*Figure 2—figure supplement 12*, *Figure 2—figure supplement 13*). These results are consistent with previous efforts demonstrating that the strongest correlations between AFDs are typically concentrated among pairs of closely related community members (i.e. low phylogenetic distance) (*Sireci et al., 2023*), implying that the effects of correlation should dissipate when communities are coarse-grained. Given that the variance of the sum can be approximated by the sum of the variances and that, by definition, the mean of a sum is the sum of the means, it is reasonable to propose that the statistical moments of coarse-grained AFDs are sufficient to characterize the distribution.

Finally, while we know of no general closed-form solution for the sum of independent gamma-distributed random variables with different rate parameters (equivalent to considering the convolution of many AFDs with different carrying capacities), progress has been made towards obtaining suitable approximations (*Stewart et al., 2007*; *Murakami, 2015*; *Hu et al., 2020*; *Behme and Bondesson, 2017*; *Barnabani, 2017*). This body of work includes an analysis demonstrating that a single gamma distribution can provide a suitable approximation to the distribution of the sum of many gamma random variables with different rate parameters (*Covo and Elalouf, 2014*). In summary, the gamma distribution appears to successfully captures patterns of biodiversity under taxonomic and phylogenetic coarse-graining because the sum of multiple gamma distributions can be approximated by a single gamma distribution.

## Predicting measures of richness and diversity within a coarse-grained scale

Given that the presence or absence of a community member is used to estimate community richness, a measure previously used to make claims about patterns of microbial diversity across taxonomic scales (*Madi et al., 2020*), we can visualize the sufficiency of the gamma distribution by predicting the mean richness within an environment at a given coarse-grained scale (*Equation 13a*). Likewise, we can use the entirety of the distribution of read counts to predict the diversity within a site, a measure that reflects richness as well as the distribution of abundances within a community (*Equation 14a*), analytic predictions that we validated through simulations (*Figure 3—figure supplement 1*). We note that we observe consistent deviations between the analytic predictions of the variance of diversity and simulation results. These deviations are likely driven by small deviations in predictions of the second moment of diversity, which are slight for individual community members, but become considerable when terms are summed over hundreds or thousands of community members.

Focusing on the human gut microbiome as an example, we found that we can predict the typical richness of a community across phylogenetic scales using the gamma distribution (*Figure 3a*). Similar results were obtained when we repeated our analysis for predicted diversity (*Figure 3b*). By examining all nine environments we found that despite the dissimilarity in environments, we were able to predict mean richness and diversity in the face of coarse-graining (*Figure 3c and d*). In contrast, the UNTB failed to predict richness (*Figure 3—figure supplement 3*). The results of this analysis suggest that the composition of microbial communities remained largely invariant under coarse-graining and that the gamma distribution remained a suitable null model for predicting mean community measures across coarse-grained scales. Identical results were obtained for taxonomic coarse-graining (*Figure 3—figure supplement 2*).

Turning to higher-order moments, we examined the variance of richness and diversity across sites. Using a similar approach that was applied to the mean, we derived analytic predictions for the variance (*Equation 17a*). With the human gut as an example, we see that analytic predictions typically fail to capture estimates of variance obtained from empirical data for phylogenetic coarse-graining (*Figure 4a and b*). This lack of predictive success was consistent across environments (*Figure 4c and d*), implying that a model of independent community members with gamma-distributed abundances

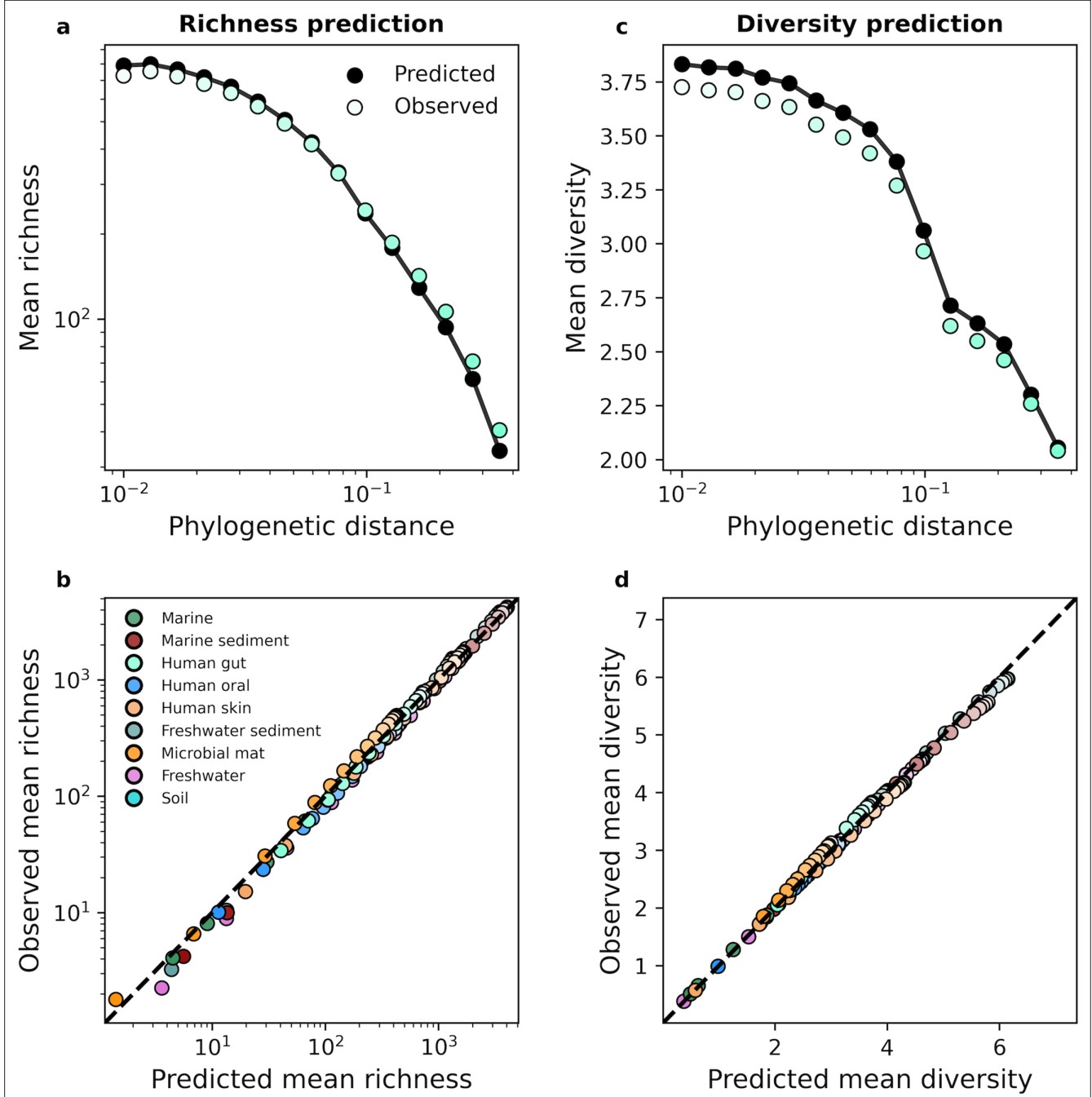

**Figure 3.** The gamma distribution successfully predicted mean richness and diversity under phylogenetic coarse-graining. (**a**) The expected richness derived from the gamma distribution (*Equation 13a*) was capable of predicting richness across phylogenetic coarse-graining scales, as illustrated by data from the human gut. (**b**) Predictions remained successful across all environments, suggesting that a minimal model of zero interactions was sufficient to predict observed properties of community composition, (**c, d**) Similarly, predictions of expected diversity (14) also succeeded across coarse-graining scales for all environments. The shade of a color of a given datapoint represents the phylogenetic distance used for coarse-graining, with lighter colors representing finer scales and darker colors representing coarser scales.

The online version of this article includes the following figure supplement(s) for figure 3:

**Figure supplement 1.** The analytic predictions of the mean and variance of richness and diversity vs. the results of simulations that assume gamma-distributed AFDs and reads drawn from a multinomial distribution.

**Figure supplement 2.** The gamma distribution successfully predicted mean richness and diversity under taxonomic coarse-graining.

**Figure supplement 3.** Unified Neutral Theory of Biodiversity (UNTB) failed to predict mean richness. UNTB consistently overpredicted richness under both taxonomic and phylogenetic coarse-graining.

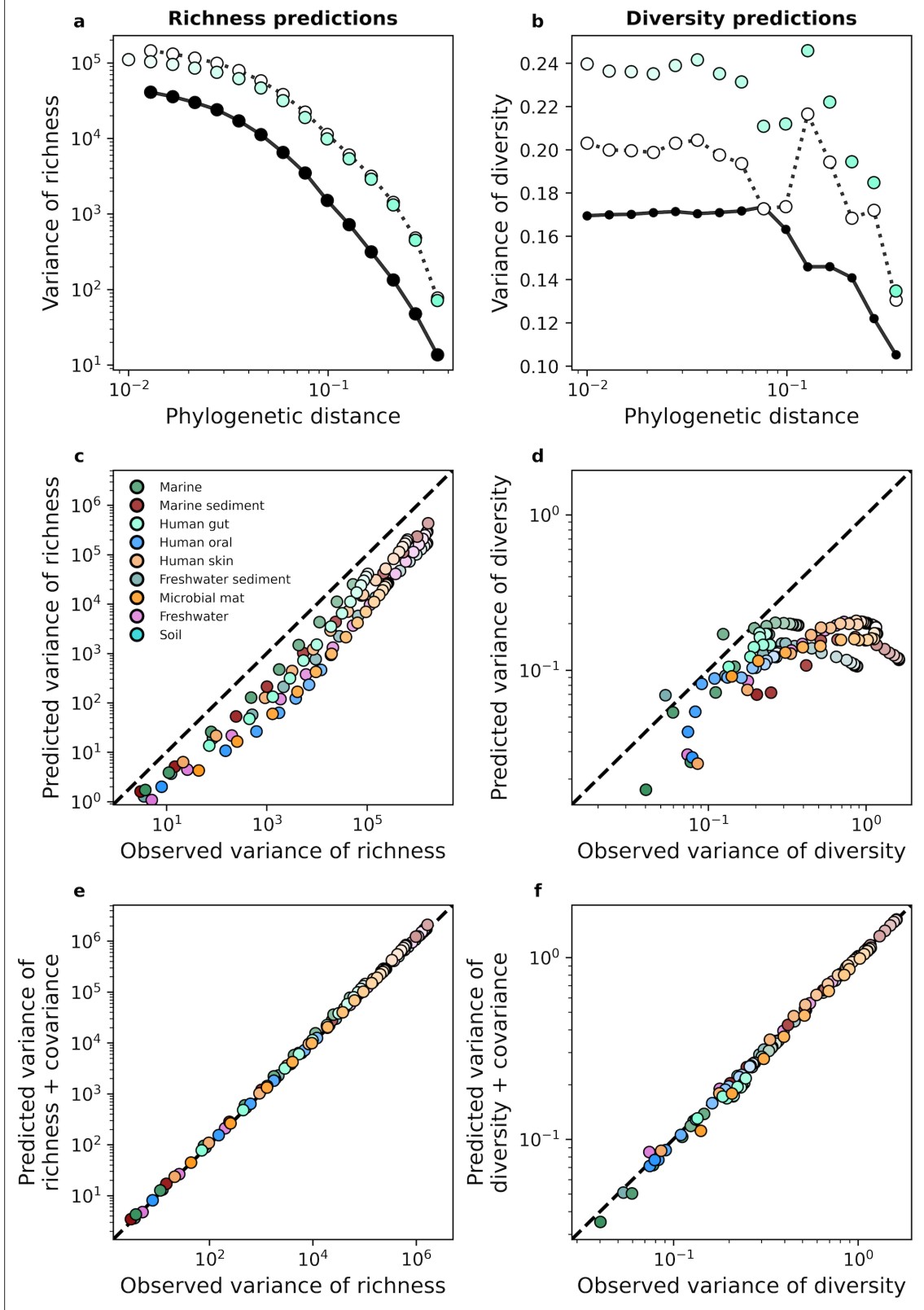

**Figure 4.** The gamma distribution only predicts the variance of richness and diversity under phylogenetic coarse-graining when covariance is included. (**a, b**) In contrast with the mean, the variance of richness and diversity estimates predicted by the gamma distribution (*Equation 17a*) failed to capture empirical estimates from the human gut. Predictions are only comparable when empirical estimates of covariance are included in the predictions of the gamma distribution, meaning that dependence among community members is essential to describe the variation in measures of biodiversity across

*Figure 4 continued on next page*

*Figure 4 continued*

communities. (**c, d**) This lack of predictive success was constant across environments, (**e, f**) though the addition of covariance consistently improves our analytic predictions. The color scale used here is identical to the color scale used in *Figure 3*.

The online version of this article includes the following figure supplement(s) for figure 4:

**Figure supplement 1.** A gamma AFD can only predict the variance of richness and diversity under taxonomic coarse-graining when covariance is included.

was insufficient to capture the variance of measures of biodiversity. A major assumption made in our derivation was that community members are independent, an assumption that is unjustified given that the gamma distribution has been previously shown to be unable to capture the empirical distribution of correlations in the AFDs of community members (*Grilli, 2020*). To attempt to remedy this failed prediction, we again turned to the law of total variance by estimating the covariance of richness and diversity from empirical data and adding the covariance to the predicted variance for each measure. We found that the addition of this empirical estimate was sufficient to predict the observed variance in the human gut (*Figure 4a and b*) as well as across environments (*Figure 4e and f*), implying that the underlying model is fundamentally correct for predicting the first moment of measures of biodiversity but cannot capture the correlations necessary to explain higher statistical moments such as the variance. Identical results were again obtained with taxonomic coarse-graining (*Figure 4—figure supplement 1*).

## Predicting patterns of richness and diversity between fine and coarse-grained scales

Our predictions of the statistical moments of richness and diversity using the gamma distribution provided the foundation necessary to investigate macroecological patterns between different taxonomic and phylogenetic scales. One such prominent pattern is the relationship between the fine-grain richness/diversity within a given coarse-grained group vs. the coarse-grained richness/diversity among all remaining groups (e.g. the number of classes within Firmicutes vs. the number of phyla excluding the phylum Firmicutes), a pattern that has been purported to demonstrate the existence of DBD processes in microbial systems. Before continuing, we note that the acronym DBD technically refers to the hypothesis that such positive relationships reflects the existence of ecological interactions through which coarse-grained diversity bolsters the accumulation of fine-grained diversity (e.g. niche construction *Laland et al., 1999*; *San Roman et al., 2018*). Since we are primarily interested in the predictive power of an empirically-validated null model of biodiversity, we distinguish between DBD as a hypothesis and DBD as an empirical pattern by referring to the slope as the fine vs. coarse-grained relationship throughout the remainder of this manuscript.

The fine vs. coarse-grained relationship can be quantified as the slope of the relationship between the fine-grained richness within a given coarse-grained group $g$ ($S_{g,m}$) and the richness in the remaining $G-1$ coarse-grained groups: $S_{g,m} \propto \alpha S_{G\backslash g,m}$, where $G \backslash g$ denotes the exclusion of group $g$ and $\alpha$ is the slope of the relationship. This formulation was proposed by Madi et al., and to ensure commensurability we adopted it here (*Madi et al., 2020*). Furthermore, keeping with the approach used by Madi et al., fine and coarse-grained measures were compared across increasing taxonomic and phylogenetic scales (e.g. OTU vs. genus, genus vs. family, etc.), (*Madi et al., 2020*). Using *Equation 1*, we then defined each of these estimators in terms of the sampling form of the gamma distribution while accounting for sampling

$$S_{g,m} = |g| - \sum_{i \in g} P(0|N_m, \bar{x}_i, \beta_i) \tag{3a}$$

$$S_{G\backslash g,m} = (|G| - 1) - \sum_{\substack{g' \in G \\ g' \neq g}} P(0|N_m, \bar{x}_{g'}, \beta_{g'}) \tag{3b}$$

Similarly, we used *Equation 14a* to derive predictions for fine and coarse-grained diversity.

$$H_{g,m} = -\sum_{i \in g} \left\langle x \ln [x] | N_m, \bar{x}_g, \beta_g \right\rangle \tag{4a}$$

$$H_{G \backslash g, m} = -\sum_{\substack{g' \in G \\ g' \neq g}} \left\langle x \ln [x] \mid N_m, \bar{x}_{g'}, \beta_{g'} \right\rangle \qquad (4b)$$

By repeating this calculation for all $M$ sites, we obtained vectors of coarse and fine-grained richness estimates for group $g$ from which we inferred the slope of the fine vs. coarse-grained relationship through ordinary least squares regression. By repeating this process for all $G$ groups we obtained a distribution of slopes that can be directly compared to those obtained from empirical data. We include a conceptual diagram visualizing this process as a supplement (*Figure 5—figure supplement 1*).

Before performing a direct comparison, we first note the features of the empirical slopes and how they pertain to the predictions we obtained. By examining the distribution of empirical slopes pooled over all coarse-graining thresholds for each environment, we found that they were rarely less than zero (*Figure 5a*, *Figure 5—figure supplement 2a*). The few negative slopes inferred from empirical data were extremely small, having absolute values $<10^{-4}$ and could be treated as zeros. Furthermore, the distribution of slopes follows the same form across environments, suggesting that the slope of the fine vs. coarse-grained relationship reflects a general feature of community sequence data rather than the ecology of specific environments. Like the empirical slopes, the gamma distribution virtually always predicted a positive slope for all environments for both taxonomic and phylogenetic coarse-graining. This paucity of negative slopes suggests that the prediction of the alternative to the DBD hypothesis, the Ecological Controls hypothesis (*Schluter and Pennell, 2017*), is virtually absent in empirical data and cannot be generated from an empirically validated null model of microbial biodiversity.

However, only observing positive slopes does not necessarily provide support for the DBD hypothesis. A direct comparison of slopes predicted from the gamma distribution to those inferred from empirical data is necessary to determine whether the predictions of DBD lie outside what can be reasonably captured by an interaction-free model such as the SLM. To evaluate the novelty of the slope of the fine vs. coarse-grained relationship we compared the values of observed slopes to those obtained from the interaction-free SLM. We found that the predictions of the gamma distribution closely matched the observed slopes across environments for both taxonomic and phylogenetic coarse-graining (*Figure 5—figure supplement 3*, *Figure 5—figure supplement 4*). We consolidated these results by taking the mean slope for a given coarse-grained scale, from which we see that the mean slope predicted by the gamma distribution does a reasonable job capturing empirical slopes across environments (*Figure 5b*, *Figure 5—figure supplement 2b*). These results indicate that we should expect to see a positive relationship between richness estimates at different scales and that the relationships we observe can be quantitatively captured by a gamma-distributed AFD. It is worth noting that the slope of the fine vs. coarse-grained relationship could be sufficiently predicted even though the gamma distribution only succeeded at predicting mean richness, suggesting that higher-order statistical moments, and by extension interactions between community members, are unnecessary to quantitatively capture the positive relationship observed between fine and coarse-grained estimates of richness.

As a point of comparison, we predicted the slope of the fine vs. coarse-grained relationship for richness using a UNTB model (*Madi et al., 2023*) (Supporting information). We found that generally, the UNTB slopes deviated from those obtained from empirical data, exhibiting far greater bias and variation around the 1:1 line than what was observed of the SLM (*Figure 5—figure supplement 5*, *Figure 5—figure supplement 6*). By examining the mean slope we found that predictions from the UNTB tended to systematically underpredict the observed slope under both taxonomic and phylogenetic coarse-graining (*Figure 5—figure supplement 7*). Directly comparing the mean relative error of the UNTB predictions to those of the SLM confirms these observations, as the UNTB predictions tended to have larger errors by an order of magnitude (*Figure 5—figure supplement 8*, *Figure 5—figure supplement 9*). To summarize, in contrast to the SLM, the UNTB cannot predict the slope of the fine vs. coarse-grained relationship for richness.

While richness is a widespread and versatile estimator that is commonly used in community ecology, neglects considerable information by focusing on presences and absences instead of the entirety of the distribution of abundances. To rigorously test the predictive power of the gamma distribution it was necessary to evaluate the fine vs. coarse-grained relationship for diversity. We again found that disparate environments had similar distributions of slopes from empirical data (*Figure 5c*, *Figure 5—figure supplement 2c*), suggesting that the slope of the relationship is likely a general property of

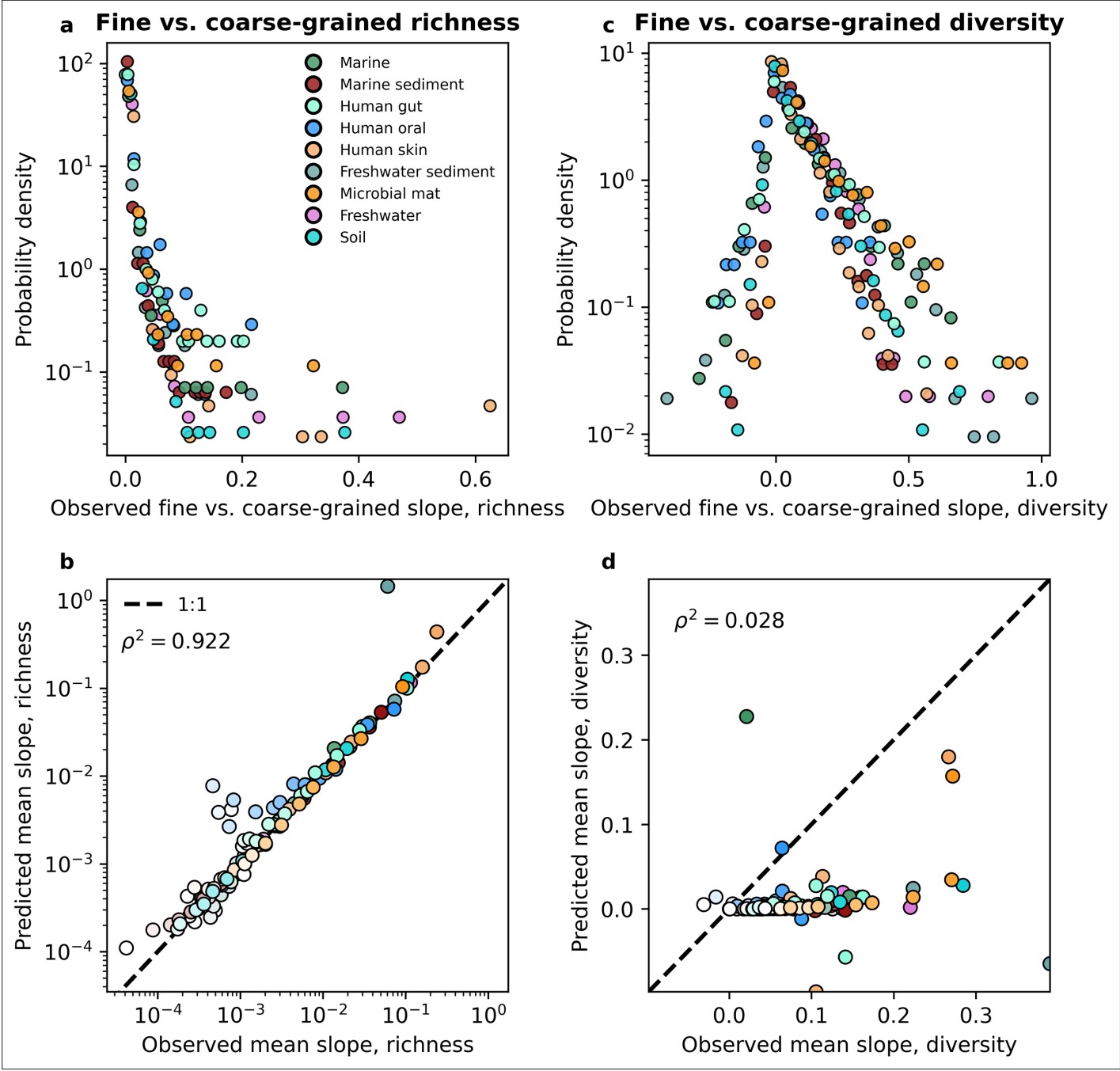

**Figure 5.** The slope of the fine vs. coarse-grained relationship for richness could be predicted by the gamma distribution, but was novel for estimates of diversity. (**a, b**) The predictions of the gamma distribution (***Equation 3a***) successfully reproduced observed fine vs. coarse-grained richness slopes across scales of phylogenetic coarse-graining. (**c, d**) In contrast, the predictions of the gamma distribution failed to capture diversity slopes (***Equation 4a***). The color scale used here is identical to the color scale used in ***Figure 3***. Squared Pearson correlation coefficients ($\rho^2$) are computed over all slopes for all taxa across all coarse-graining scales.

The online version of this article includes the following figure supplement(s) for figure 5:

**Figure supplement 1.** Conceptual diagram illustrating how fine vs. coarse-grained slopes are inferred.

**Figure supplement 2.** The gamma distribution as a tool for investigating the novelty of fine vs.coarse-grained slopes.

**Figure supplement 3.** The predicted slopes of fine vs. coarse-grained richness from the sampling form of the gamma distribution under taxonomic coarse-graining.

*Figure 5 continued on next page*

*Figure 5 continued*

**Figure supplement 4.** The predicted slopes of fine vs. coarse-grained richness from the sampling form of the gamma distribution under phylogenetic coarse-graining.

**Figure supplement 5.** The predicted slopes of fine vs. coarse-grained richness under taxonomic coarse-graining using the UNTB.

**Figure supplement 6.** The predicted slopes of fine vs. coarse-grained richness under phylogenetic coarse-graining using the UNTB.

**Figure supplement 7.** The mean predicted slopes of fine vs. coarse-grained richness under (**a**) taxonomic and (**b**) phylogenetic coarse-graining using the Unified Neutral Theory of Biodiversity (UNTB).

**Figure supplement 8.** Comparisons of the relative error of fine vs. coarse-grained richness slope predictions between the Stochastic Logistic Model (SLM) and Unified Neutral Theory of Biodiversity (UNTB) for taxonomic coarse-graining.

**Figure supplement 9.** Comparisons of the relative error of fine vs. coarse-grained richness slope predictions between the Stochastic Logistic Model (SLM) and Unified Neutral Theory of Biodiversity (UNTB) for phylogenetic coarse-graining.

**Figure supplement 10.** The predicted slopes of fine vs. coarse-grained diversity from the sampling form of the gamma distribution under taxonomic coarse-graining.

**Figure supplement 11.** The predicted slopes of fine vs. coarse-grained diversity from the sampling form of the gamma distribution under phylogenetic coarse-graining.

microbial communities rather than an environment-specific pattern. However, unlike richness, diversity predictions obtained from the gamma distribution generally failed to capture observed slopes, as the squared correlation between observed and predicted slopes can be less than that of richness by over an order of magnitude (*Figure 5d*, *Figure 5—figure supplement 10*, *Figure 5—figure supplement 11*, *Figure 5—figure supplement 2d*). Here, we see where the predictions of an interaction-free SLM succeeded and failed to predict observed macroecological patterns.

Given that the gamma distribution failed to predict the observed diversity slope, it is worth evaluating whether additional features could be incorporated to generate successful predictions. A notable omission is that there is an absence of interactions between community members in the SLM, meaning that we were unable to predict correlations between community member abundances. However, while considerable progress has been made (e.g. *Ho et al., 2022*), predicting the observed distribution of correlation coefficients between community members while accounting for sampling remains a non-trivial task. Given that the gamma distribution succeeded at predicting other macroecological patterns, we elected to perform a simulation where a collection of sites was modeled as an ensemble of communities with correlated gamma-distributed AFDs with the means, variances, correlations, and total depth of sampling set by estimates from empirical data (Materials and methods). By including correlations between AFDs into the simulations, the statistical outcome of ecological interactions between community members, we were able to largely capture observed fine vs. coarse-grained diversity slopes (*Figure 6*, *Figure 6—figure supplement 1*, *Figure 6—figure supplement 2*, *Figure 6—figure supplement 3*). These results suggest that rather than diversity at a fine-scale begetting diversity at a coarse-scale, the correlations that exist at a fine-scale (e.g. genus) contribute to measures of biodiversity at the nearest coarse-grained scale (e.g. family), resulting in a positive relationship between measures of diversity at different scales.

## Discussion

The results of this study demonstrate that macroecological patterns in microbial communities remain largely invariant across taxonomic and phylogenetic scales. By focusing on the predictions of the SLM, an interaction-free model of microbial growth under environmental fluctuations, we were able to evaluate the extent that measures of biodiversity can be predicted under coarse-graining. We were largely able to predict said measures using the same model with parameters estimated from data across scales, implying that certain macroecological patterns of microbial communities remained self-similar across taxonomic and phylogenetic scales. Building off of this result, we investigated the dependence of community measures between different degrees of coarse-graining, a pattern that has been formalized as the Diversity Begets Diversity hypothesis (*Whittaker, 1972*; *Madi et al., 2020*). The prediction derived from the sampling form of the gamma distribution quantitatively captured the observed slopes of the fine vs. coarse-grained relationship for richness, while it failed to capture

the slope of diversity. However, introducing correlations between abundance fluctuation distribution permitted the recovery of the slope of the fine vs. coarse-grained diversity relationship.

Our richness results complement past work demonstrating that occupancy, the constituent of richness, is highly dependent on two parameters: sampling depth (i.e. total read count) and the mean abundance of a community member (*Grilli, 2020*). Our ability to predict the relationship between fine and coarse-grained measures of richness using the gamma distribution, despite our inability to predict the variance of richness, suggest that correlations driving the slope of the fine vs. coarse-grained relationship is primarily driven by the effects of finite sampling. This past work, and the relationships between the mean abundance and occupancy evaluated in this manuscript, demonstrate that occupancy alone is unlikely to contain ecological information that is not already captured by the distribution of abundances across sites (i.e. the AFD). Our analyses of the relationship between fine and coarse-grained richness support this conclusion, as predictions derived from a gamma distribution quantitatively captured the observed slope. The success of an interaction-free model in predicting the slope of the fine vs. coarse-grained relationship is an indictment of the appropriateness of estimators that rely solely on the presence of a community member for identifying novel macroecological patterns, a measure that has been used to bolster support for the DBD hypothesis at the level of 16S rRNA amplicons as well as strains (*Madi et al., 2020*; *Madi et al., 2023*). Rather, estimates of richness harbor little information about the dynamics of a community across taxonomic and phylogenetic scales that is not already captured by the sampling form of the gamma distribution. Contrasting with richness, the predictions of diversity from the gamma distribution were unable to capture fine vs. coarse-grained relationships in empirical data. Given that measures of diversity incorporate information about the richness and evenness of a community (*Magurran, 2004*), the comparative deficiency of our predictions for fine vs. coarse-grained diversity suggests that forms of the SLM that neglect interactions between community members cannot capture relationships between phylogenetic/taxonomic scales that depend on the evenness of the distribution of abundances.

Macroecological patterns are not imbued with mechanistic explanation (*Warren et al., 2022*). Rather, the onus is on the investigator to identify plausible mechanisms. Often in ecology this task is made easier by evaluating whether a model lacking a particular mechanism is capable of producing the observed pattern, that is, identifying an appropriate null. The novelty of the fine vs. coarse-grained relationship was previously assessed using a null model which assumed demographic equivalence among community members and community dynamics driven by demographic noise (i.e. the UNTB) (*Madi et al., 2020*; *Alonso and McKane, 2004*). Empirical patterns of microbial abundance cannot be reasonably captured by such models, making predictions obtained from the UNTB invalid for evaluating the novelty of microbial macroecological patterns. In contrast, models that combine self-limiting growth with environmental noise reproduces several empirical patterns, making the SLM an appropriate choice for evaluating the novelty of fine vs. coarse-grained relationships (*Grilli, 2020*; *Descheemaeker and de Buyl, 2020*). This is not a trivial detail, as there is historical precedence on the need to identify an appropriate null in order to investigate how fine and coarse-grained measures of biodiversity relate to one another, as one of the earliest adoptions of null model analysis in ecology was done to investigate the ratio of species to genera in a community (*Williams, 1947*; *Smith et al., 2014*).

In this study, the predictions of the sampling form of the gamma distribution considerably improved when correlations between community members were included. This result suggests that rather than exclusively pointing to niche construction as previously suggested (*Madi et al., 2020*), any ecological mechanism that can capture the observed distribution of correlation coefficients is a plausible candidate. Given that models of consumer-resource dynamics have succeeded in capturing macroecological patterns (*Chesson, 1990*; *Cui et al., 2021*), including quantitatively predicting the distribution of correlation coefficients (*Ho et al., 2022*), it is reasonable to suggest that such mechanisms are ultimately responsible for the relationship between fine and coarse-grained measures of diversity and can be reduced to phenomenological models such as the SLM. Indeed, experimental investigations of the slopes evaluated here have found the existence of positive slopes in artificial communities maintained in a laboratory setting, where the strength of the correlation between fine and coarse-grained scales is driven by the secretion of secondary metabolites (*Estrela et al., 2022*). This mechanism, known as cross-feeding, can be viewed as compatible with the concept of niche construction (*San Roman et al., 2018*) as well as with the original interpretation of *Madi et al., 2020*.

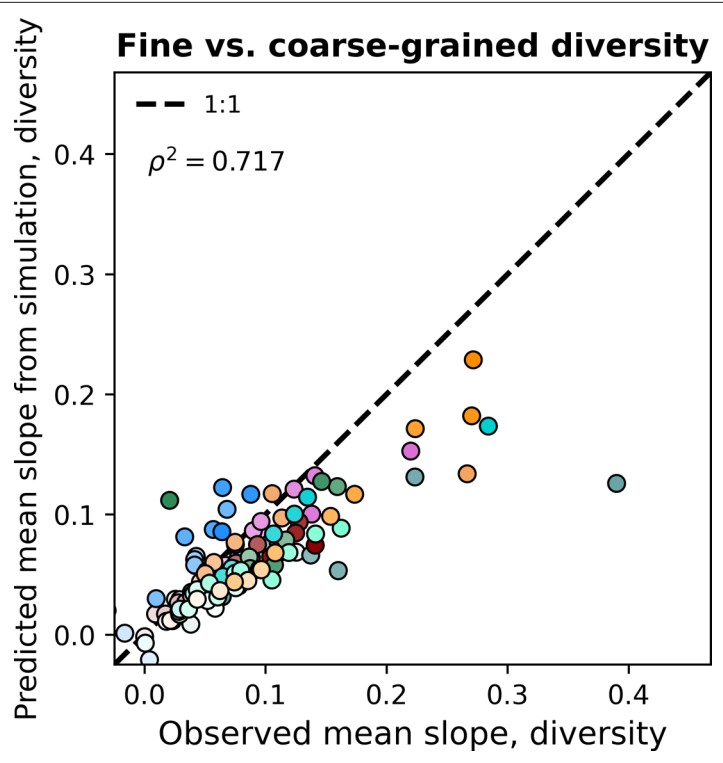

**Figure 6.** Including correlations allows the gamma distribution to capture observed diversity slopes. Observed fine vs. coarse-grained diversity slopes could be quantitatively reproduced under phylogenetic coarse-graining by simulating correlated gamma-distributed AFDs at the OTU-level. The color scale used here is identical to the color scale used in **Figure 3**. Squared Pearson correlation coefficients ($\rho^2$) are computed over all slopes for all taxa across all coarse-graining scales.

The online version of this article includes the following figure supplement(s) for figure 6:

**Figure supplement 1.** The predicted slopes of fine vs. coarse-grained diversity from the sampling form of the gamma distribution with correlations between OTUs under taxonomic coarse-graining.

**Figure supplement 2.** The predicted slopes of fine vs. coarse-grained diversity from the sampling form of the gamma distribution with correlations between OTUs under phylogenetic coarse-graining.

**Figure supplement 3.** Gamma distribution simulations with correlations capture observed diversity slopes.

In the interest of providing macroecological insight into the DBD hypothesis, we solely focused on coarse-graining procedures that relied on phylogenetic reconstruction and taxonomic assignment. However, it is worth noting that it is also possible to coarse-grain community members by the strength of their correlations (i.e. sum the abundances of each pair of community members with the strongest correlation in AFDs). This procedure has been named the phenomenological renormalization group method due to its ability to identify if and where a system is stable despite knowing little about the system's dynamics (i.e. fixed points in nonlinear systems) (**Nicoletti et al., 2020**; **Meshulam et al., 2019**). However, given that the AFD correlation between two community members is often inversely related to their phylogenetic distance, such an analysis would likely be redundant, as coarse-graining based on the strength of correlation would effectively coarse-grain the most closely related community members (**Sireci et al., 2023**).

A major goal of this study was to evaluate the novelty of macroecological patterns that were used to bolster support for the DBD hypothesis. We used the same dataset in order to ensure generality and commensurability with past research efforts. However, it is worth inspecting how the use of a global survey dataset constrains the inferences one can make. Throughout this study, we implicitly assumed that an ensemble approach is valid, meaning that we viewed different sites/hosts as virtual copies of a given environment. This assumption can remain valid for time-series studies where the distribution of microbial abundances remains stationary with respect to time (**Faith et al., 2013**), as the stationary

**Table 1.** Summary statistics for the 100 sites randomly selected for each environment. These statistics reflect the data used for taxonomic coarse-graining, as OTUs lacking taxonomic labels were excluded from taxonomic coarse-graining analyses.

| Environment | Total # OTUs | Mean # OTUs | Mean # reads |
|---|---|---|---|
| Marine | 9090 | 690.80 | 95,129.43 |
| Marine sediment | 16,110 | 1,393.67 | 40,440.38 |
| Human gut | 6175 | 599.09 | 32,894.50 |
| Human oral | 4716 | 537.45 | 44,271.22 |
| Human skin | 17,955 | 1293.45 | 36,344.13 |
| Freshwater sediment | 12,231 | 1080.95 | 18,979.56 |
| Microbial mat | 5087 | 200.24 | 8,659.36 |
| Freshwater | 12,052 | 822.37 | 33,646.53 |
| Soil | 20,298 | 1814.76 | 36,268.93 |

solution of the SLM has successfully characterized microbial community time-series at both the level of OTUs (*Grilli, 2020*) and strains (*Wolff et al., 2023*). Given these past results, we predict that the fine vs. coarse-grained relationship results presented here will remain valid in longitudinal studies where community members fluctuate around a single point with respect to time.

# Materials and methods
## Data acquisition and processing

To ensure that our analyses were generalizable across ecosystems and commensurate with prior DBD investigations, we used amplicon sequence data from the V4 region of the 16S rRNA gene generated and curated by the Earth Microbiome Project (*Thompson et al., 2017*; *Madi et al., 2020*). We restricted our analysis to the quality control (QC)-filtered subset of the EMP, which was annotated using the closed-reference database SILVA (*Quast et al., 2013*) and consists of 96 studies culminating in 23,828 total samples with each processed sample having ≥10,000 reads. We downloaded the public Silva reference tree for OTUs with 97% similarity 97_otus.tre from the EMP database. We identified nine heavily sampled environments in the metadata file emp_qiime_mapping_qc_filtered. tsv and selected 100 random sites from each environment. Summary statistics for each environment are provided (*Table 1*, *Table 2*).

We briefly note that our occupancy and richness predictions depend on the form of the gamma distribution that explicitly accounts for sampling as a multinomial process. The multinomial distribution

**Table 2.** Summary statistics for the 100 sites randomly selected for each environment. These statistics reflect the data used for phylogenetic coarse-graining as all OTUs could be used.

| Environment | Total # OTUs | Mean # OTUs | Mean # Reads |
|---|---|---|---|
| Marine | 18,173 | 1,356.37 | 168,520.66 |
| Marine sediment | 41,304 | 4,167.25 | 106,166.92 |
| Human gut | 10,190 | 862.73 | 44,031.12 |
| Human oral | 7062 | 614.97 | 46,104.84 |
| Human skin | 29,448 | 1817.12 | 48,285.68 |
| Freshwater sediment | 33,193 | 3,569.59 | 65,582.56 |
| Microbial mat | 11,869 | 431.42 | 23,216.02 |
| Freshwater | 26,645 | 1775.89 | 74,298.17 |
| Soil | 45,273 | 4730.74 | 106,578.45 |

describes the probability of sampling $n$ reads given a relative abundance of $x$ and total read count $N$ with replacement, a process we can model as the Poisson limit of a binomial sampling process for individual community members. Given this choice and the past success of the gamma distribution, we deviated from past analyses by electing to not sub-sample read counts to the same depth, as the process of sampling without a replacement would bias the sampling distribution for rare community members (*Madi et al., 2020*).

## Coarse-graining protocol

Taxonomic coarse-graining was performed as the summation of the abundances of all OTUs within a given taxonomic group. We removed taxa with indeterminate labels to prevent potential biases due to taxonomic misassignment, (e.g. 'uncultured,' 'ambiguous taxa,' 'candidatus,' 'unclassified,' etc.). Manual inspection of EMP taxonomic annotations revealed a low number of OTUs that had been assigned the taxonomic label of their host (e.g. *Arachis hypogaea* (peanut)). These marked OTUs were removed from all downstream analyses.

Phylogenetic coarse-graining was performed using the phylogenetic tree provided by SILVA 123 97_otus.tre in the EMP release. Each internal node of a phylogenetic tree was collapsed if the mean branch lengths of its descendants was less than a given distance. All phylogenetic operations were performed using the Python package ETE3 (*Huerta-Cepas et al., 2016*).

## Deriving biodiversity measure predictions

While the gamma distribution as the stationary solution of the SLM and the sampling form of the gamma distribution have been previously derived (*Grilli, 2020*), we briefly outline relevant derivations here for the convenience of the reader before deriving the predicted richness and diversity of a community. We define the SLM as the following Langevin equation

$$\frac{dx_i}{dt} = \underbrace{\frac{x_i}{\tau_i}\left(1 - \frac{x_i}{K_i}\right)}_{\text{Self-limiting growth}} + \underbrace{\sqrt{\frac{\sigma_{\tau_i}}{\tau_i}}x_i \cdot \eta(t)}_{\text{Environmental noise}}$$

(5)

Here $\tau_i$, $K_i$, and $\sigma_{\tau_i}$ represent the timescale of growth, the carrying capacity, and the coefficient of variation of growth rate fluctuations, respectively. Multiplicative environmental noise is captured by the product of a linear frequency term, the coefficient of variation of growth rate fluctuations, and a Brownian noise term $\eta(t)$ that introduces stochasticity into the equation. The expected value of $\eta(t)$ is $\langle\eta(t)\rangle = 0$ (*Gardiner, 2009*). The dependence of $\eta(t')$ at time $t'$ on an earlier time $\eta(t)$ is defined as $\langle\eta(t)\eta(t')\rangle = \delta(t - t')$(*Gardiner, 2009*). This standard definition means that if the noise term is shifted in time, it has zero correlation with itself. We briefly note that because DBD patterns were originally investigated by Madi et al., using an ensemble of sites that belong to the same type of ecosystem rather than the time series of a single site (*Madi et al., 2020*), the gamma distribution alone does not prove the validity of the SLM nor does it prove alternatively formulated stochastic differential equations of ecology that also predict a gamma distribution (e.g. *George and O'Dwyer, 2022*). However, given that the SLM has successfully characterized the temporal dynamics of microbial communities, we believe that this model is an appropriate formulation for investigating DBD patterns (*Grilli, 2020*; *Wolff et al., 2023*; *Descheemaeker and de Buyl, 2020*).

In contrast to the SLM, macroecological predictions can be derived from the UNTB. There are many forms of the UNTB, but the novelty of observed fine vs. coarse-grained relationships was assessed using a form of the UNTB that predicts that the distribution of community member abundances within a given site follows a zero-sum multinomial distribution (*Alonso and McKane, 2004*; *Madi et al., 2020*). For the convenience of the reader the predicted richness using the form of the UNTB relevant to this study has been rederived (Supporting information).

The stationary distribution of the SLM can be derived using the Itô ↔ Fokker-Planck equivalence and solving for the stationary solution (*Grilli, 2020*; *Engen and Lande, 1996*), resulting in the gamma-distributed AFD. Through the SLM, we can define the mean relative abundance and its squared inverse coefficient of variation as $\bar{x}_i = K_i\left(1 - \frac{\sigma_{\tau_i}}{2}\right)$ and $\beta_i = \frac{2-\sigma_{\tau_i}}{\sigma_{\tau_i}}$, respectively. These are parameters that were estimated from the empirical data and were used below to obtain predictions. Using these definitions and the stationary distribution of *Equation 5*, we obtained the gamma distribution

$$P(x_i|\bar{x}_i, \beta_i) = \frac{1}{\Gamma(\beta_i)} \left(\frac{\beta_i}{\bar{x}_i}\right)^{\beta_i} \exp\left[-x_i \frac{\beta_i}{\bar{x}_i}\right] x_i^{\beta_i-1} \tag{6}$$

When we sequence microbial communities, one obtains read counts rather than actual abundances. Therefore, it is necessary to account for the reality of sampling when we apply to empirical data. We can account for sampling by first assuming that the probability of observing a single community member can be modeled as a binomial sampling process. Given that the total number of reads is typically large ($N \gg 1$) and the typical relative abundance of a community member is much smaller than one ($x_i \ll 1$), the binomial can be approximated as a Poisson sampling process with the following probability of sampling $n$ reads

$$P(n|N, x_i) = \frac{(N \cdot x_i)^n e^{-N \cdot x_i}}{n!} \tag{7}$$

This formulation of the sampling process is convenient, as it can be used to obtain an analytic solution for the probability of observing $n$ reads given $\bar{x}_i$ and $\beta_i$, the parameters we estimate from the data. This distribution can be obtained by solving the convolution of the Poisson and the gamma distribution (**Grilli, 2020**). The resulting distribution can be considered a negative binomial distribution if sites have identical sampling depths (**Fisher, 1941**). Using this distribution, we calculated the probability of obtaining $n_m$ reads out of a total sampling depth of $N_m$ for the $i$th OTU in sample $m$ as

$$P(n_m|N_m, \bar{x}_i, \beta_i) = \int_0^\infty P(x_i|\bar{x}_i, \beta_i) \cdot P(nn_m|N_m, x_i) dx_i \tag{8a}$$

$$= \frac{\Gamma(\beta_i + n_m)}{n_m! \Gamma(\beta_i)} \left(\frac{\bar{x}_i N_m}{\beta_i + \bar{x}_i N_m}\right)^{n_m} \left(\frac{\beta_i}{\beta_i + \bar{x}_i N_m}\right)^{\beta_i} \tag{8b}$$

This distribution requires two parameters that can be estimated from the data ($\bar{x}_i$ and $\beta_i$) and one parameter that is known (total number of reads, $N_m$). This equation will be used to obtain predictions of measures of biodiversity. First, noticing that the probability of a community member's absence is the complement of its presence, we can define the expected occupancy of a community member across $M$ sites as

$$\langle o_i \rangle = \frac{1}{M} \sum_m^M \left(1 - P(0|N_m, \bar{x}_i, \beta_i)\right) \tag{9}$$

And the second moment of occupancy as

$$\langle o_i^2 \rangle = \frac{1}{M} \sum_m^M \left(1 - P(0|N_m, \bar{x}_i, \beta_i)\right)^2 \tag{10}$$

from which we defined the predicted variance of occupancy

$$\mathrm{Var}(o_i) = \langle o_i^2 \rangle - \langle o_i \rangle^2 \tag{11}$$

The success of our predictions was assessed using the relative error.

$$\varepsilon = \left|\frac{\mathrm{Obs.} - \mathrm{Pred.}}{\mathrm{Obs.}}\right| \tag{12}$$

Using the definition of occupancy from the sampling form of the gamma distribution, we derived the expected richness of a community as

$$\langle S \rangle = \sum_{i=1}^{S_{\text{total}}} \langle o_i \rangle \tag{13a}$$

$$= \frac{1}{M} \sum_{m=1}^M \sum_{i=1}^{S_{\text{obs}}} (1 - P(0|N_m, \bar{x}_i, \beta_i)) \tag{13b}$$

$$= S_{\text{total}} - \frac{1}{M}\sum_{m=1}^{M}\sum_{i=1}^{S_{\text{total}}} P(0|N_m, \bar{x}_i, \beta_i) \tag{13c}$$

where $S_{\text{total}}$ is the total number of observed community members. Similarly, we derived the expected value of Shannon's diversity (**Magurran, 2004**).

$$\langle H \rangle = \frac{1}{M}\sum_{m=1}^{M}\langle H_m \rangle \tag{14a}$$

$$= -\frac{1}{M}\sum_{m=1}^{M}\sum_{i=1}^{S_{\text{obs}}}\left\langle x\ln\left[x\right]|N_m, \bar{x}_i, \beta_i\right\rangle \tag{14b}$$

$$= -\frac{1}{M}\sum_{m=1}^{M}\sum_{i=1}^{S_{\text{total}}}\int_0^{N_m}\frac{n'}{N_m}\ln\left[\frac{n'}{N_m}\right]\cdot P(n'|N_m, \bar{x}_i, \beta_i)dn' \tag{14c}$$

In physics parlance, these predictions neglect interactions between community members, also known as mean-field predictions. We then calculated the mean-field prediction of *Equation 13a* from empirical data. However, there is no known analytic solution for the integral inside the sum of *Equation 14a*. To calculate $\langle H \rangle$, we performed numerical integration on each integral for each taxon in each sample at a given coarse-grained resolution using the quad() function from SciPy.

To predict the variance of each measure we derived the expected value of the second moment, assuming independence among community members. We derived the second moments of richness and diversity.

$$\left\langle S^2 \right\rangle = \left\langle \left(\sum_i^{S_{\text{total}}} o_i\right)^2 \right\rangle \tag{15a}$$

$$= \frac{1}{M}\sum_m^{M}\left(\sum_i^{S_{\text{total}}} o_{i,m}\right)^2 \tag{15b}$$

$$= \frac{1}{M}\sum_m^{M}\sum_{i,j} o_{i,m}o_{j,m}\underbrace{\int dn P(n|N, \bar{x}_i, \beta_i)\delta_{N,N_m}}_{=1} \tag{15c}$$

$$= \int dn P(n|N, \bar{x}_i, \beta_i)\sum_{i,j}\frac{1}{M}\sum_m^{M}\delta_{N,N_m} o_{i,m}o_{j,m} \tag{15d}$$

$$= \sum_{i=1}^{S_{\text{total}}}\int dn P(n|N, \bar{x}_i, \beta_i)o_i^2\delta_{N,N_m} + \sum_{i\neq j}\int dn P(n|N, \bar{x}_i, \beta_i)o_i o_j\delta_{N,N_m} \tag{15e}$$

$$= \sum_{i=1}^{S_{\text{total}}}\left\langle o_i^2|N_m, \bar{x}_i, \beta_i\right\rangle + \sum_{i\neq j}\left\langle o_i|N_m, \bar{x}_i, \beta_i\right\rangle\left\langle o_j|N_m, \bar{x}_j, \beta_j\right\rangle \tag{15f}$$

where $\delta_{i,j}$ is the Kronecker delta.

By performing an analogous series of operations, we obtained the expected value of the second moment for diversity.

$$\left\langle H^2 \right\rangle = \left\langle \left(\sum_i^{S_{\text{total}}} x_i\ln\left[x_i\right]\right)^2 \right\rangle \tag{16a}$$

$$= \frac{1}{M}\sum_m^{M}\left(\sum_i^{S_{\text{total}}} x_i\ln\left[x_i\right]\right)^2 \tag{16b}$$

$$= \frac{1}{M}\sum_m^{M}\sum_{i,j}\left(x_i\ln\left[x_i\right]\right)\left(x_j\ln\left[x_j\right]\right)\cdot\int dn P(n|N, \bar{x}_i, \beta_i)\delta_{N,N_m} \tag{16c}$$

$$= \frac{1}{M} \sum_m^M \sum_{i=1}^{S_{\text{total}}} \int dn P(n|N, \bar{x}_i, \beta_i) \delta_{N,N_m} (x_i \ln [x_i])^2 \tag{16d}$$

$$+ \frac{1}{M} \sum_m^M \sum_{i \neq j} \int dn P(n|N, \bar{x}_i, \beta_i) \delta_{N,N_m} (x_i \ln [x_i])(x_j \ln [x_j]) \tag{16e}$$

$$= \frac{1}{M} \sum_m^M \sum_{i=1}^{S_{\text{total}}} \left\langle (x \ln [x])^2 | N_m, \bar{x}_i, \beta_i \right\rangle \tag{16f}$$

$$+ \frac{1}{M} \sum_{i \neq j} \sum_m^M \left\langle x \ln [x] | N_m, \bar{x}_i, \beta_i \right\rangle \left\langle x \ln [x] | N_m, \bar{x}_j, \beta_j \right\rangle \tag{16g}$$

Where the expected value of the second moment of the diversity term is defined as $\left\langle (x \ln [x])^2 | N_m, \bar{x}_s, \beta_s \right\rangle = \int_0^{N_m} \left( \frac{n'}{N_m} \ln \left[ \frac{n'}{N_m} \right] \right)^2 \cdot P(n'|N_m, \bar{x}_s, \beta_s) dn'$. From which we obtained the expected value of the variance

$$\text{Var}(S) = \left\langle S^2 \right\rangle - \left\langle S \right\rangle^2 \tag{17a}$$

$$\text{Var}(H) = \left\langle H^2 \right\rangle - \left\langle H \right\rangle^2 \tag{17b}$$

We predicted the mean and variance of richness and diversity separately at each coarse-grained scale. Specifically, we coarse-grained the empirical data, estimate $\bar{x}_s$ and $\beta_s$ for each coarse-grained community member, and use these estimates to obtain a prediction for each measure of biodiversity.

It is worth noting why the above functions constitute predictions. To obtain values that we can compare with empirical data we estimated the mean and variance of relative abundance across sites for each community member at a given scale. These parameters were used to obtain the expected value of a community-level measure (e.g. richness) using a function. These functions were derived under the assumption that a given probability distribution (i.e. the gamma) provided an appropriate description of the distribution of relative abundances across sites. We then compared the expected value of a community-level measure to the mean value from empirical data and assessed the similarity between the two values.

### Fine vs. coarse-grained relationship slope inference

In order to predict the relationship between the measures within a coarse-grained group and that among all remaining groups, we calculated a vector of predicted richness or diversity estimates for all sites using *Equation 3a* or *Equation 4a* within a given coarse-grained group and 3b or *Equation 4b* among the remaining groups. This 'leave-one-out' procedure was originally implemented by Madi et al., where the authors examined the slope of fine vs. coarse-grained measures of diversity as a sliding window across taxonomic ranks with both the fine and coarse scales increasing with each rank (e.g. genus:family, family: order, etc.) (*Madi et al., 2020*). To maintain consistency, we used the same definition for our predictions. We also extended the definition to the case of phylogenetic coarse-graining, where we compared fine and coarse scales using different phylogenetic distances while retaining the same ratio (e.g. 0.1:0.3, 0.3:0.5, etc.). Slopes were estimated using ordinary least squares regression with SciPy. Throughout the manuscript the success of a prediction was evaluated by calculating its relative error as follows: we only inferred the slope if a fine-grained group had at least five members. We only examined the slopes of a given coarse-grained threshold if at least three slopes could be inferred.

### Simulating communities of correlated gamma-distributed AFDs

Correlated gamma-distributed AFDs were simulated by performing inverse transform sampling. For each environment with $M$ sites, an $M \times S_{\text{obs}}$ matrix $Z$ was generated from the standard Gaussian distribution using the empirical $S_{\text{obs}} \times S_{\text{obs}}$ correlation matrix calculated from relative abundances. The cumulative distribution $U = \Phi(Z)_{Gaus.}$ was calculated and a matrix of the abundances of community members across sites was obtained using the point percentile function of the gamma distribution and

the empirical distribution of mean relative abundances and the squared inverse coefficient of variation of abundances: $\bar{x} = \bar{x}_1, \bar{x}_2, \cdots, \bar{x}_{S_{obs}}$, $\boldsymbol{\beta} = \beta_1, \beta_2, \cdots, \beta_{S_{obs}}$. To simulate the process of sampling, each community of the resulting $M \times S_{obs}$ matrix of true relative abundances $X = \Phi(U)_{Gamma}^{-1}$ was sampled using a multinomial distribution with the empirical distribution of total read counts.

## Acknowledgements

This work was supported by the NSF Postdoctoral Research Fellowships in Biology Program under Grant No. 2010885 (WRS).

## Additional information

### Funding

| Funder | Grant reference number | Author |
|---|---|---|
| National Science Foundation | 2010885 | William R Shoemaker |

The funders had no role in study design, data collection and interpretation, or the decision to submit the work for publication.

### Author contributions

William R Shoemaker, Conceptualization, Resources, Data curation, Software, Formal analysis, Funding acquisition, Validation, Investigation, Visualization, Methodology, Writing – original draft, Project administration, Writing – review and editing; Jacopo Grilli, Conceptualization, Formal analysis, Supervision, Methodology, Writing – original draft, Writing – review and editing

### Author ORCIDs

William R Shoemaker (ID) https://orcid.org/0000-0003-0111-4838

Reviewer #1 (Public Review): https://doi.org/10.7554/eLife.89650.3.sa1
Reviewer #3 (Public Review): https://doi.org/10.7554/eLife.89650.3.sa2
Author Response https://doi.org/10.7554/eLife.89650.3.sa3

## Additional files

### Supplementary files

• MDAR checklist

### Data availability

All sequencing data used in this study was obtained from the Earth Microbiome Project (URL: https://ftp.microbio.me/emp/release1/). Processed data used to perform the analyses in this study are available on Zenodo, DOI: https://doi.org/10.5281/zenodo.7692046. All code written for this study is available on GitHub under a GNU General Public License: https://github.com/wrshoemaker/macroeco_phylo (copy archived at *Shoemaker, 2023b*).

The following dataset was generated:

| Author(s) | Year | Dataset title | Dataset URL | Database and Identifier |
|---|---|---|---|---|
| Shoemaker WR | 2023 | Macroecological patterns in coarse-grained microbial communities | https://doi.org/10.5281/zenodo.7692046 | Zenodo, 10.5281/zenodo.7692046 |

The following previously published datasets were used:

| Author(s) | Year | Dataset title | Dataset URL | Database and Identifier |
|---|---|---|---|---|
| Thompson LR, Sanders JG, McDonald D, Amir A, Ladau J, Locey KJ, Prill RJ, Tripathi A, Gibbons SM, Ackermann G, Navas-Molina JA, Janssen S, Kopylova E, Vázquez-Baeza Y, González A, Morton JT, Mirarab S, Xu ZZ, Jiang L, Haroon MF, Kanbar J, Zhu Q, Song SJ, Kosciolek T, The Earth Microbiome Project Consortium | 2017 | Earth Microbiome Project mapping files | https://ftp.microbio.me/emp/release1/mapping_files/ | Earth Microbiome Project, release1/mapping_files |
| Thompson LR, Sanders JG, McDonald D, Amir A, Ladau J, Locey KJ, Prill RJ, Tripathi A, Gibbons SM, Ackermann G, Navas-Molina JA, Janssen S, Kopylova E, Vázquez-Baeza Y, González A, Morton JT, Mirarab S, Xu ZZ, Jiang L, Haroon MF, Kanbar J, Zhu Q, Song SJ, Kosciolek T, The Earth Microbiome Project Consortium | 2017 | Earth Microbiome Project phylogeny and taxonomy | https://ftp.microbio.me/emp/release1/otu_info/silva_123/ | Earth Microbiome Project, release1/otu_info/silva_123 |
| Thompson LR, Sanders JG, McDonald D, Amir A, Ladau J, Locey KJ, Prill RJ, Tripathi A, Gibbons SM, Ackermann G, Navas-Molina JA, Janssen S, Kopylova E, Vázquez-Baeza Y, González A, Morton JT, Mirarab S, Xu ZZ, Jiang L, Haroon MF, Kanbar J, Zhu Q, Song SJ, Kosciolek T, The Earth Microbiome Project Consortium | 2017 | Earth Microbiome Project count data | https://ftp.microbio.me/emp/release1/otu_tables/closed_ref_silva/ | Earth Microbiome Project, release1/otu_tables/closed_ref_silva |

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

## Appendix 1

### Supporting information: Investigating macroecological patterns in coarse-grained microbial communities using the stochastic logistic model of growth

#### UNTB richness predictions

Below we rederive a prediction for richness using the form of the UNTB used by Madi et al., as a point of comparison to the SLM predictions derived in the main manuscript (*Madi et al., 2020*; *Alonso and McKane, 2004*). When the size of a metacommunity tends towards an asymptotic limit, the stationary distribution for community members of relative abundance $x$ approaches the following continuous distribution (*Vallade and Houchmandzadeh, 2003*).

$$P(x|\theta)dx = \frac{\theta}{x}(1-x)^{\theta-1}dx \tag{S1}$$

where $\theta$ is Hubbell's biodiversity parameter (also known as Fisher's $\alpha$ *Fisher et al., 1943*).

Using this distribution, we can obtain an expression for the expected number of community members with $n$ sampled individuals out of a total sample size of $N$.

$$S(n|N, m, \theta) = \theta \int_0^1 P(n|N, m, x)\frac{(1-x)^{\theta-1}}{x}dx \tag{S2}$$

where $P(n; N, m, x)$ is the probability of sampling $n$ individuals of relative abundance $x$ given a total sample size $N$

$$P(n|N, m, x) = \binom{N}{n} \frac{\Gamma(n+\gamma x)}{\Gamma(\gamma x)} \frac{\Gamma(N+\gamma(1-x)-n)}{\Gamma(\gamma(1-x))} \frac{\Gamma(\gamma)}{\Gamma(\gamma+N)} \tag{S3}$$

where $\gamma = \frac{m(N-1)}{1-m}$. The function *Equation S2* is known as the migration-limited zero-sum multinomial distribution (ZSM) (*Alonso and McKane, 2004*). As $m \to 1$, *Equation S2* approaches a limiting form known as the metacommunity zero-sum multinomial distribution (mZSM). The process of sampling community members under the mZSM can be represented as a binomial distribution.

$$S(n|N, \theta) = \theta \int_0^1 x^N(1-x)^{N-n}\frac{(1-x)^{\theta-1}}{x}dx \tag{S4}$$

Similar to our analysis using the SLM, the binomial can be approximated as a Poisson distribution.

$$S(n|N, \theta) = \theta \int_0^1 e^{-xN}\frac{(xN)^n}{n!}\frac{(1-x)^{\theta-1}}{x}dx \tag{S5}$$

The resulting integral can be obtained by using the change of variable $y = xN$ and rearranging terms, then approximating the upper limit of integration as infinity (since $N \gg 1$).

$$S(n|N, \theta) = \theta \int_0^1 e^{-y}\frac{(y)^n}{n!}\left(1-\frac{y}{N}\right)^{\theta-1}\frac{N}{y}\frac{dy}{N} \tag{S6a}$$

$$= \frac{\theta}{n}\int_0^1 \underbrace{e^{-y}\frac{y^{n-1}}{(n-1)!}}_{\text{Gamma distribution}}\left(1-\frac{y}{N}\right)^{\theta-1}dy \tag{S6b}$$

$$\approx \frac{\theta}{n}\int_0^\infty \underbrace{e^{-y}\frac{y^{n-1}}{(n-1)!}}_{\text{Gamma distribution}}\left(1-\frac{y}{N}\right)^{\theta-1}dy \tag{S6c}$$

$$= \frac{\theta}{n}\left\langle \left(1-\frac{Y}{N}\right)^{\theta-1}\right\rangle \tag{S6d}$$

By rearranging terms, we obtain a gamma distribution with shape parameter $n$ and rate parameter 1. Because we integrated over a product with a gamma distribution, the variable $Y$ is a gamma-distributed random variable. We can then expand term in the integral using a Taylor series around $Y = n$ and by noticing that $\langle Y \rangle = n$ and $\langle Y^2 \rangle = n^2 + n$ under a gamma distribution.

$$S(n|N, \theta) = \frac{\theta}{n} \left(1 - \frac{n}{N}\right)^{\theta-1} + \frac{1}{2} \frac{\theta(\theta-1)(\theta-2)}{N^2} \left(1 - \frac{n}{N}\right)^{\theta-3} + \mathcal{O}\left(N^{-3}\right) \tag{S7}$$

We can then predict the richness of a community by summing the abundances from 1 to $N$.

$$S(N, \theta) = \sum_{n=1}^{N} S(n|N, \theta) \tag{S8}$$

This quantity represents the total observed richness of a sample from a panmictic infinite metacommunity after accounting for sampling. Predictions of mean richness over $M$ sites can then be calculated as

$$\langle S(\theta) \rangle = \frac{1}{M} \sum_{m=1}^{M} S(N_m, \theta) \tag{S9}$$

## Simulating fine vs. coarse-grained richness slopes under the UNTB

We followed the procedure in Madi et al., to obtain fine vs. coarse-grained slopes for richness so that they could be compared to predictions obtained from the SLM (*Madi et al., 2020*). We simulated SADs according to the mZSM model outlined above using the rmzsm() function from the R package sads v0.4.2. We simulated 100 SADs using the empirical distribution of total read counts and the total number of observed OTUs. We set the biodiversity parameter $\theta = 50$ for all environments. The SADs returned by rmzsm() contain no zeros, meaning that values of richness are identical for all UNTB SADs. In order to introduce zeros so that richness estimates could vary, we followed the procedure used in Madi et al., where each simulated SAD was rarefied to 5000 individuals. We repeated this rarefaction procedure on the empirical SADs. We then performed taxonomic and phylogenetic coarse-graining and fine vs. coarse-grained slope inference using the procedure described in the Materials and methods.

