## [Editor Report · eLife assessment]

This **valuable** study considers empirical macroecological patterns in microbiome data across multiple taxonomic scales. The work **convincingly** shows that the Stochastic Logistic Growth model is a more appropriate choice of null model than the neutral theory of biodiversity. The work will be of particular interest to microbial ecologists.

---

## [Referee Report · Reviewer #1 (Public Review)]

Shoemaker and Grilli analyze publicly available sequencing data to quantify how the microbial diversity of ecosystems changes with the taxonomic scale considered (e.g., diversity of genera vs diversity of families). This study builds directly on Grilli's 2020 paper which used this data to show that for many different microbial species, the distribution of abundances of the species across sampling sites belongs to a simple one-parameter family of gamma distributions. In this work, they show that the gamma distribution also describes the distribution of abundances of higher taxonomic levels. The distribution now requires two parameters, but the second parameter can be approximately derived by treating the distributions of lower-level taxonomic units as being independent. The difference between the species-level result and the result at higher taxonomic levels suggests that in some sense microbial species are ecologically meaningful units.

While the higher-level taxon abundance distributions can be well-approximated assuming independence of the constituent species, this approach substantially underestimates variation in community richness and diversity among sampling sites. Much of this extra variability appears to be driven by variability in sample size across sites. It is not clear to me how much this variation in sample size is itself due to variation in sampling effort versus variation in overall microbial densities. This variation in sample size also produces correlations between taxon richness at lower and higher taxonomic levels. For instance, sites with large samples are likely to have both many species within a genus and many genera. The authors also consider taxon diversity (Shannon index, i.e. entropy), which is constructed from frequencies and is therefore less sensitive to sample size. In this case, correlations between diversity across taxonomic scales instead appear to depend on the idiosyncratic correlations among species abundances.

This paper's results are presented in a fairly terse manner, even when they are describing summary statistics that require a lot of thought to interpret. I don't think it would make sense to try to understand it without having first worked through the 2020 paper. But everyone interested in a general understanding of microbial ecology should read the 2020 paper, and once one has done that, this paper is worth reading as well simply for seeing how the major pattern in that paper shifts as one moves up in taxonomic scale.

---

## [Referee Report · Reviewer #3 (Public Review)]

Summary

In this research advance, the authors purport to show that the unified neutral theory of biodiversity (UNTB) is not a suitable null model for exploring the relationship between macroecological quantities, and additionally that the stochastic logistic growth model (SLM) is a viable replacement. They do this by citing other studies where UNTB was unable to capture individual macroecological quantities, and then demonstrating SLM's strength at predicting the same diversity metrics. They extend this analysis to show SLM's modeling capability at multiple scales of coarse graining, in addition to its failures at predicting these metrics' variances. Finally, authors conduct a similar analysis to Madi et al. (2020) by investigating the relationship between diversity measures within a group and across coarse-grained groups (e.g. genera diversity in one family compared to diversity of families). The authors show that choosing SLM as a null model reveals some previously reported relationships to be no longer "novel", in the sense that the patterns can be adequately captured by the null model. Authors also show that relationships not captured by the null model can be recovered by adding correlations, suggesting interactions are the driving force behind them.

Strengths

1. Authors make a strong argument that UNTB is not a good null model of macroecological observables and especially relationships between them. Authors convincingly argue that a SLM is a better null since the gamma distribution it predicts is a better description of the empirical Abundance Fluctuation Distributions (AFD).

2. Authors show that the gamma distribution predicted by SLM is a good fit for the AFD's at many different scales of coarse graining, not just the OTU level as was previously demonstrated. Authors show the same distribution predicted the mean diversity and richness at all scales of coarse graining.

3. Authors convincingly demonstrate how SLM can be used to test the relevance of interactions to macroecological relationships.

Weaknesses

This reviewer's concerns were convincingly addressed by the revisions.

Overall Impact

The authors present a convincing argument for the use of SLM as a better non-interacting null model for macroecological quantities and relationships.

---

## [Author Response]

The following is the authors’ response to the original reviews.

**Reviewer #1:**
In no particular order:1. In Figs S3 and S4, can they also show gamma fit? (or rather corrected fit accounting for abundance conditioning?) The shapes look different, especially for the microbial mat.

Author response: We have added gamma distribution fits to the rescaled AFD plots (Figs. S3, S4).

1. Lines 170-176 seem like they should come before lines 164-166.

Author response: In lines 166-170 we discuss empirical patterns in the data that motivate the introduction of the SLM as a model in lines 170-175. We have clarified these points in the revision.

1. The wiggles in the gamma predictions in the occupancy-abundance plots are because occupancy depends not only on abundance but also on the shape parameter, right? Probably good to write a sentence or two explaining what's going on here.

Author response: We agree with the reviewer that the variation in the prediction could be in-part driven by variation in the shape parameter across community members. We now include this observation in our revision (lines 209-211).

1. In the predicted vs observed occupancy plots, it would be nice to add curves showing predicted standard deviation or similar to give a sense of how well the model is predicting the variability.

Author response: In the revised manuscript we now include predictions for the variance of occupancy using the gamma distribution under both taxonomic and phylogenetic coarse-graining (Fig. S9; S10; lines 211-214).

1. Covariance between sister groups: Figs S9 and S10 look very nice, but it's hard to see much because they're log-log plots over multiple decades, while even a several-fold difference from y = x would indicate a strong effect of correlations. It would be clearer if the y-axis showed the ratio of the coarsegrained variance to the sum of OTU variances and we were looking at how well it fit y = 1.

Author response: We have included these plots in the revision (Fig. S14, S15).

1. If the sum of gammas can be well-approximated by a gamma, does that mean that the gamma is just a fairly flexible distribution and we shouldn't take the quality of the gamma fits in general as a very specific indication of what's going on?

Author response: While the sum of random variables that are drawn from gamma distributions with different parameters is often well-approximated by another gamma, this does not tell us why the gamma distribution holds for microbial communities at the finest-grain level (i.e.,OTUs/ASVs). At present, the best explanation is that the gamma is a stationary distribution for certain stochastic differential equations which have ecological interpretations (Grilli, 2020; Shoemaker et al., 2023). Furthermore, alternative two-parameter distributions have been tested alongside the gamma and have done a comparatively poor job capturing observed macroecological patterns (Grilli, 2020). These results suggest that the utility of the gamma distribution is not simply an outcome of its flexible nature, it succeeds because it has captured core ecological properties of microbial communities. In the case of the SLM, gamma-like distributions arise when a community member is subject to self-limiting growth and environmental noise. On the other hand, the stability of the gamma distribution might explain why it can be detected as shape of the AFD, as it does not fade out across coarse-graining level.

1. What's going on with the variance of diversity in Fig S12? Does this suggest that some of the problem in Figure 4 could be with the analytic approximation rather than the model? I had a hard time understanding the part of the Methods explaining the simulation details (lines 587-597). It would be worth expanding this. Is there some way to explain how the correlations were simulated in terms of the SLM, e.g., correlations in the noise term across OTUs?

Author response: We believe that deviations in the variance of diversity in Fig. S16g,h are driven by small deviations in our predictions of the second moment (x∗ln(x)|Nm,x¯i,βi2) (Eq. S16). Alone these predictions are slight, but their effects become noticeable when summed over hundreds or thousands of taxa. We have included this observation in the revised manuscript (lines 268-271). However, this deviation pales in comparison with the magnitude of covariance in the empirical data, suggesting that our inability to predict the variance of richness and diversity is primarily driven by our assumption of statistical independence.

Regarding the source of the correlations, under the SLM correlations in abundances can be introduced either by adding deterministic interaction terms or through correlated environmental noise. Determining which of these two options drives empirical correlations is an active area of research (e.g., Camacho-Mateu et al., 2023). For the purpose of this study, we remain agnostic on the cause of the correlations, optioning to instead emphasize that that the inclusion of correlations is necessary to reproduce observed slopes of the fine vs. coarse-grained relationship for diversity.

1. In Figure 5ab, is the idea that the correlation in richness is primarily driven by the number of samples from the environment? Line 390 seems to say so, but it would be good to make this explicit and put it right in that section of the Results.

Author response: Our results suggest that sampling effort (# reads) plays a larger role in determining the correlations between fine and coarse-grained measures of richness. We now clarify this point in the revised manuscript (lines 429-435).

1. I don't totally understand the contrast in lines 369-372. If fine-scale diversity within one group begets coarse-grained diversity in another group, couldn't that show up as correlations in the AFDs? Or is the argument that only including within-group correlations in AFDs is enough to reproduce the pattern? I'm not sure I see how that could be.

Author response: The term “begets” implies both causation and direction. If we see a positive relationship between diversity estimates at two different scales of observation the causal mechanism cannot be determined solely from correlations between samples obtained once from different sites. So, mechanisms consistent with niche construction/"DBD" can produce correlations, though the existence of correlations do not necessarily imply DBD.

1. The discussion of niche construction on 429-431 doesn't match very well with 440-441. Basically, niche construction is a very broad concept, not a specific one, right?

Author response: In lines 472-576 (formerly 429-431) we discuss how the existence of correlations between fine and coarse-grained scales does not point to a single ecological mechanism. Alternatively stated, observing a non-zero slope does not mean that niche construction is driving the relationship.

In lines 476-487 (formerly 440-441) we discuss how the mechanism of cross-feeding has been shown to generate a positive relationship between fine and coarse-grained measures of diversity. This mechanism can be interpreted as a form of “niche construction”, so it is an instance of a tested ecological mechanism that aligns with the interpretation given in Madi et al. (2020).

1. Isn't (8) just the negative binomial distribution?

Author response: The convolution of the stationary solution of the SLM (i.e., a gamma distribution) and the Poisson limit of a multinomial sampling distribution returns a negative binomial distribution of read counts across hosts if samples have identical sampling depths. We now include this detail in the revision (line 593-595). Note however that if different samples have different sampling depths, the distribution of reads across samples is not a negative binomial.

1. Missing 1/M in (9).

Author response: We have fixed this omission in the revision.

1. Schematic figures illustrating what the different statistics are intuitively capturing would really help this work be understandable to a broader audience, but they'd also be a ton of work.

Author response: Richness and diversity are used in ecology to such an extent that we do not see the benefit of a conceptual diagram. Furthermore, we have included a conceptual diagram about our pipeline in our revision at the request of Reviewer 2 (Fig. S20).

**Reviewer #2:**
Major RecommendationsIf I were reviewing this manuscript for a regular journal, I believe the following issues would be important to address prior to publication.1. From my reading, the main points of this advance are thata. SLM models AFDs well at all levels of coarse-graining.b. This makes SLM a better null-model than UNTB for macroecological relationships.c. Using SLM on the EMP data, the richness slopes are well explained by SLM but not the diversity slopes. Therefore, any theory that hopes to explain the diversity slopes must include interactions. Argument B appears to be one of the key points yet is missing from the abstract, and should be made clearer. If these aren't the main points the authors intended, then other main points need to be highlighted more.

Author response: In the revision we now explicitly mention argument b in the Abstract.

1. The title should be more specific, so as to better reflect the content. (E.g. "UNTB is not a good null model for macroecological patterns" would seem more appropriate.)

Author response: We would prefer to focus on the success of the SLM rather than the limitations of the UNTB in the title of this work. Therefore, we have modified our title as follows: “Investigating macroecological patterns in coarse-grained microbial communities using the stochastic logistic model of growth”.

1. The manuscript would benefit from a clearer description of exactly what information the SLM retains about the data (perhaps even a cartoon panel in one of the figures). In particular, it is important to be explicit about the number of model parameters.

Author response: The number of model parameters for the gamma AFD are now explicitly stated in the revision (Lines 579-580).

1. The main point of Figures 2-4 seems to be that SLM is good at describing the data (and when it fails it is due to interactions) while UNTB fails to reproduce this behavior, in support of Argument B. This is not clear from the figure descriptions or titles, which focus on SLM's "predictive" power.

Author response: Fig. 2a demonstrates that the gamma distribution predicted by the SLM explains the empirical distribution of abundances. This result provides motivation to predict the fraction of sites harboring a given community member (i.e., occupancy, Fig. 2c) as well as general measures of community composition including mean richness (Fig. 3a,c) and mean diversity (Fig. 3b,d) using parameters estimated from the data (not free parameters).

This success led us to consider whether the gamma distribution could predict the variance of richness and diversity, which it could not because it does not capture covariance between community members (Fig. 4).

In the revision we have identified opportunities to make these points clear throughout the Results.Furthermore, we have added additional detail to the legends of Figs. 2-4.

1. The manuscript would benefit from clarifying the use of "prediction" related to the SLM. Since the gamma distributions predicted by SLM were fit to empirical data, it seems like the agreement between analytic means and empirical means (Fig. 3) is a statement on gamma distributions being a good fit for the AFD's more than SLM predicting richness and diversity. For example, from my reading, it seems like this analysis could be done numerically by shuffling species abundances across environments and seeing whether this changed the mean richness/diversity. I would not call this shuffling test a prediction, since it is more a statement on the relevance of interactions. SLM predicts gamma-distributed AFD's, but those distributions recovering the data they were trained on doesn't seem like a prediction.

Author response: In this manuscript we identified the gamma distribution as an appropriate probability distribution to describe the distribution of relative abundances across samples over a range of coarse-grained scales. Motivated by this result, we performed a separate analysis where at each scale we estimated the mean and variance of relative abundance across sites for each community member. We then used these parameters to obtain the expected value of acommunity-level measure using an equation we derived by assuming that the gamma distribution was appropriate (e.g., richness, Eq. 13). We then compared the expected value of richness to the mean value from empirical data and assessed the similarity between the two values.

The outcome of this procedure constitutes a prediction. While the mean and variance are parameters, estimating them from the empirical data has no connection with the operation of training a distribution on empirical data. We could have derived predictions such as Eq. 13 using any other probability distribution that can be parameterized using the mean and variance (e.g., Gaussian). Such a prediction would likely do a poor job even though it used the same means and variances used for our gamma predictions. This is because the choice of distribution would not have been a good descriptor of the distribution of abundances across hosts.

To better explain this last -- perhaps the most significant -- issue, I'd like to ask the authors if the following recasting would be an accurate reflection of their conclusions, or if something is missing.1. "Focusing on the empirical relationship observed between diversity slopes by Madi 2020, we ask the question: does explaining these relationships require accounting for species-species correlations? Or could it be reproduced in a noninteracting model?"To address this question, one can perform a randomization test, shuffling abundances to preserve all single-OTU statistics but breaking any correlations. My reading of the authors' results is that (**new result 1**) the richness relationships would be preserved, while diversity relationships would not be preserved. [Note that this result 1 need not mention either SLM or UNTB.]

Author response: The question of whether correlations between species are necessary to explain the observed slope of the fine vs. coarse-grained relationship was only one component of our research goals. Our first question was whether the SLM would prove to be a more appropriate null for evaluating the novelty of observed slopes. We believe that our results support the conclusion that the SLM is an appropriate null for this question, as it was able to capture observed slopes of the fine vs. coarse-grained relationship for estimates of richness, determining that correlations and the interactions that are ultimately responsible are not necessary to explain this result.

We then find that the SLM as a null model fails to capture observed slopes of the fine vs. coarsegrained relationship for estimates of diversity and simulate the SLM with correlations to return reasonable estimates of the slope. However, here the question about correlations is a direct follow-up from our question about a null model that excludes interactions, so it is unclear how a randomization test would relate to this result.

1. Instead of doing a randomization test (resampling the empirical distribution), one might insist on instead fitting a model to the AFD distributions, and sampling from that distribution rather than the empirical one.a. If doing it this way, one should of course ensure that the distribution being fit is a good description of the data.b. UNTB is a bad fit. SLM is a better fit, and in fact (**new result 2**) continues to be a good empirical fit even at coarse-grained levels.c. Can make statements on using SLM as a null model for these types of cross-scale relationships. Could try arguing that fitting an SLM model per-OTU (instead of resampling the empirical distribution) could offer some advantage if certain properties could be computed analytically from the fit parameters, instead of averaging over multiple computational rounds of resampling.Do these two points accurately summarize the manuscript? If so, this presentation avoids the confusion with "prediction". If my summary is missing some important point, the presentation should be revised to clarify the points I appear to have missed.

Author response: In our manuscript we derive predictions from the gamma distribution, the stationary distribution of the SLM, that require parameters estimated from the data (i.e., mean and variance of relative abundance). These parameters are estimated from the data using normal procedures and then plugged into our predictions that assume the appropriateness of the gamma, returning values that are then compared to estimates from empirical data. Our estimation of the mean and variance does not assume that the empirical distribution following a gamma distribution, but the value returned by our function derived from the gamma distribution (e.g., Eq. 13) does make that assumption.

To address the reviewer’s broader comment, we believe that following points summarize our manuscript:

1. The gamma distribution as a stationary solution of the SLM captures macroecological patterns and predicts typical community-level properties (i.e., mean richness and diversity) across phylogenetic and taxonomic scales.

2. The gamma distribution fails to predict variation in community-level properties (i.e., variance of richness and diversity) across phylogenetic and taxonomic scales. This occurs because the SLM is a mean-field model that does not explicitly include interactions between community members.

3. Despite the inability to capture interactions, the gamma distribution succeeds at predicting the fine vs. coarse-grain slope for richness, a pattern that had previously been attributed to community member interactions. This result demonstrates that the novelty of a macroecological pattern hinges on one’s choice of null model.

4. However, the gamma cannot capture the same relationship for diversity. Simulations of the gamma distribution that incorporate correlations between community members are capable of generating reasonable estimates of the slope.

To address the reviewer’s comments regarding the appropriateness fitted gamma distributions, in our revision we have added fitted gamma distributions to plots of AFDs so that the reader can visually assess the ability of the gamma to describe empirical patterns (Fig. S3, S4).

We have also obtained predictions for the slope of the fine vs. coarse-grained relationship for community richness using the same form of UNTB used by Madi et al (2020). In our revised manuscript we establish a procedure to infer the single parameter of this model, generate predictions of richness at fine and coarse-grained scales, and then evaluate whether the UNTB is capable of predicting the slope of the fine vs. coarse-grained relationship for richness (Supplementary Information; Figs. S18, 24-28; lines 277-278; 370-380).

Other/minor comments1. The manuscript would be improved with more consistent terminology ("fine vs. coarse-grained relationship"/"the relationship" vs. "diversity slope"). Also, many readers may be used to OTUs referring to the rather fine level of description, as opposed to any chosen level; and could interpret indexing over groups as being in contrast with indexing over OTU's (coarse vs fine). The authors' use is perfectly correct, but keeping a consistent terminology would help.

Author response: We have revised our manuscript to specify the “slope” as the “slope of the fine vs. coarse-grained relationship” (e.g., Line 318). We also specify in the Results and in the Methods that we use “fine” and “coarse” as relative terms, keeping with the sliding-scale approach used in Madi et al (2020).

1. While I appreciate this "slope" is something borrowed from other work, the clarity of the paper might benefit from a cartoon of how one goes from the raw data to the slopes at a particular coarse-graining level. (Optional).

Author response: We had added a conceptual diagram to the revision (Fig. S20).

1. The text often colloquially references "the gamma," "predictions of the gamma," etc. This phrasing comes across as sloppy, and the manuscript would be improved by being more specific.

Author response: We now specify “gamma” as the “gamma distribution” throughout the manuscript.

1. Equation 6 appears to be missing some subscripts on the x terms (included on the left of the equation).

Author response: We thank the reviewer for noticing this error and we have corrected it in the revision.

1. In "Simulating communities of correlated...AFDs", the acronym SAD is not defined.

Author response: We thank the reviewer for noticing this error and we have corrected it in the revision.

1. In Figure 2:a. Invariant is probably the wrong word for the title, since all the AFD's were rescaled by mean and variance before being compared. Data does support that the gamma distributions are good at describing the AFD's, but as stated in the description it's the general shape that is preserved, not the distribution itself.

Author response: When we mention the invariance of the AFD we now specify that we mean that the shape of the distribution remained qualitatively invariant.

b. I'd recommend changing the color coding to something with more contrast, since currently it's impossible to assess the claim that the shape of the distribution collapses.

Author response: Our coarse-graining procedure is a sequential operation that has no intuitive point that would suggest the use of a contrasting colormap (e.g., if our scale ranged from -1 to 1 then there would be a natural point of contrast at zero).

c. The legend is missing relevant technical details: How many OTU's were used to make plot a? How many samples?

Author response: The number of samples was listed in the Materials and Methods (line 523). In the revision we now include a table with the average and total number of OTUs as well as the average number of reads for each environment (Table S1, S2).

d. In plot b, is the mean relative abundance referring to "mean abundance when observed" or "mean across all samples"?

Author response: The mean relative abundance is the mean abundance across all sites (line 204) and in the legend of Fig. 2.

e. Since one argument here is that SLM fits these distributions better than UNTB, if possible it would be nice to see UNTB's failed fits here.

Author response: A major feature of the UNTB is that the demographic parameters of community members are indistinguishable. Under the SLM, the variation in the mean relative abundance we observe suggests that the carrying capacities of community members vary over multiple orders of magnitude, a result that is incompatible with most forms of the UNTB (x-axis of Fig. 2b). We now mention this point in the revised manuscript (lines 110; 229; 455-471).

1. In Figure 3:a. It is not clear how coarse-graining is included in model fitting. The "Deriving biodiversity measure predictions" section would benefit from including how coarse-graining is incorporated.

Author response: We predict measures of biodiversity separately at each coarse-grained scale. We now clarify this detail in the revised manuscript (Lines 624-627).

b. Reference Shannon Diversity in Methods.

Author response: We now cite Shannon’s diversity.

c. What is the blue/white color coding in plots a & c? It doesn't have any color key.

Author response: Figs. 3-6 use a uniform light-to-dark scale for all environments, with each environment having its own color. For example, Fig. 3a contains data from the human gut microbiome. Human gut data were assigned the color aquamarine, so the shade of aquamarine for a given datapoint in Fig. 3a indicates the phylogenetic scale.

In the revision we now clarify the colorscale in the legend of Fig. 3 and specify that the same scale is used in all subsequent figure legends.

d. Re: earlier comments, why is richness considered a prediction? (Am I correct in my interpretation that panel b is almost a tautology - counting the number of zeros in the matrix either by rows or by columns - whereas panel d is nontrivial?)

Author response: Mean richness as a measure of biodiversity depends on the fraction of sites where a given community member is present (i.e., occupancy). The mean relative abundance of a community member and its variation across sites (beta) is clearly related to occupancy, but those two statistics do not give you a prediction of occupancy. Obtaining a prediction of occupancy and, subsequently, richness, requires 1) a probability distribution of abundances (i.e., the gamma) and 2) a probability distribution of sampling (i.e., the Poisson). Using these two pieces of information, we derived a prediction for mean richness (Eq. 13). We then compare the value of richness obtained by plugging in the mean relative abundances, betas, and known number of reads to the observed mean richness obtained from the data.

e. The lettering of subplots in Figure 3 is not consistent with Figure 4. Figure 3 subplots are also cited incorrectly in paragraph two on page six (lines 251-254).

Author response: We thank the reviewer for noticing the error and we have corrected it in the revision.

f. Again, if possible show UNTB predictions in plots a & c.

Author response: In our revised manuscript we provide extensive descriptions and predictions of mean richness and the slope of the fine vs. coarse-grained relationship for richness using the form of the UNTB used in Madi et al. (2020; Figs. S18, S24 - S29; lines 277-282; 370-380). We then compare the error of these slope predictions to those obtained from the SLM, finding that the SLM generally outperforms UNTB (Figs. S27-S29).

1. In Figure 4:a. What are the color codings in plots a & b?

Author response: The color scale used in Fig. 4 is identical to the color scale used in Fig. 3. This detail is now specified in the legend of Fig. 4.

b. What are the two lines of empirical data in plots a & b, and why is one of them dashed?

Author response: We now specify what the two lines mean in the key within the figure.

c. Same comment as earlier on predictions and richness.

Author response: We now specify what the two lines mean in the key within the figure.

1. In Figure 5:a. It wasn't clear to me in the manuscript how the authors generated these plots from the raw data. The manuscript would benefit from a clear cartoon/description of the data pipeline, from raw data to empirical (and analytic) slopes.

Author response: We have added a conceptual diagram to the revised manuscript (Fig. S20).

b. Make the figure title more descriptive to better connect it to the figure's objective (the richness slopes relationship is not novel, but the diversity slopes relationship is).

Author response: We have revised the figure title.

References

Camacho-Mateu, J., Lampo, A., Sireci, M., Muñoz, M. Á., & Cuesta, J. A. (2023). Species interactions reproduce abundance correlations patterns in microbial communities(arXiv:2305.19154). arXiv. https://doi.org/10.48550/arXiv.2305.19154

Grilli, J. (2020). Macroecological laws describe variation and diversity in microbial communities. Nature Communications, 11(1), 4743. https://doi.org/10.1038/s41467-020-18529-y

Madi, N., Vos, M., Murall, C. L., Legendre, P., & Shapiro, B. J. (2020). Does diversity beget diversity in microbiomes? eLife, 9, e58999. https://doi.org/10.7554/eLife.58999

Shoemaker, W. R., Sánchez, Á., & Grilli, J. (2023). Macroecological laws in experimental microbial systems (p. 2023.07.24.550281). bioRxiv. https://doi.org/10.1101/2023.07.24.550281